# Mixtures of Locally Bounded Langevin dynamics for Bayesian Model Averaging

**Dr Kilian Zepf***
*Technical University of Denmark, Kongens Lyngby, Denmark*                    *kmze@dtu.dk*

**Dr Tareen Dawood***
*Technical University of Denmark, Kongens Lyngby, Denmark*                    *tarda@dtu.dk*

**Prof. Aasa Feragen**
*Technical University of Denmark, Kongens Lyngby, Denmark*                    *afhar@dtu.dk*

**Prof. Ender Konukoglu**                                         *kender@vision.ee.ethz.ch*
*ETH Zürich, Biomedizinische Bildverarbeitung, Switzerland*

*\*Shared First Authorship*

**Reviewed on OpenReview:** *https://openreview.net/forum?id=ibqfadKjgo*

## Abstract

Properties of probability distributions change when going from low to high dimensions, to the extent that they exhibit counterintuitive behavior. Gaussian distributions intuitively illustrate a well-known effect of moving to higher dimensions, namely that the typical set almost surely does not contain the mean, which is the distribution's most probable point. This can be problematic in Bayesian Deep Learning, as the samples drawn from the high-dimensional posterior distribution are often used as Monte Carlo samples to estimate the integral of the predictive distribution. Here, the predictive distribution will reflect the behavior of the samples and, therefore, of the typical set. For instance, we cannot expect to sample networks close to the maximum a posteriori estimate after fitting a Gaussian approximation to the posterior using the Laplace method. In this paper, we introduce a method that aims to mitigate this typicality problem in high dimensions by sampling from the posterior with Langevin dynamics on a restricted support enforced by a reflective boundary condition. We demonstrate how this leads to improved posterior estimates by illustrating its capacity for fine-grained out-of-distribution (OOD) ranking on the Morpho-MNIST dataset.

## 1 Introduction

Epistemic uncertainty in machine learning captures a model's lack of knowledge and represents how unsure the model is about its own predictions. This type of uncertainty can be reduced by providing the model with more diverse and informative training data (Gawlikowski et al., 2021). It plays a central role in identifying samples for which the model's predictions cannot be trusted, including both OOD inputs and challenging in-distribution (ID) cases. Even more so, an accurate estimation of epistemic uncertainty can play a crucial role in identifying domain shifts.

To this end, given a finite sample size and an overparameterized neural network model, which is often the case, identifying different sets of "optimal weights" is key to estimating epistemic uncertainty of the model.

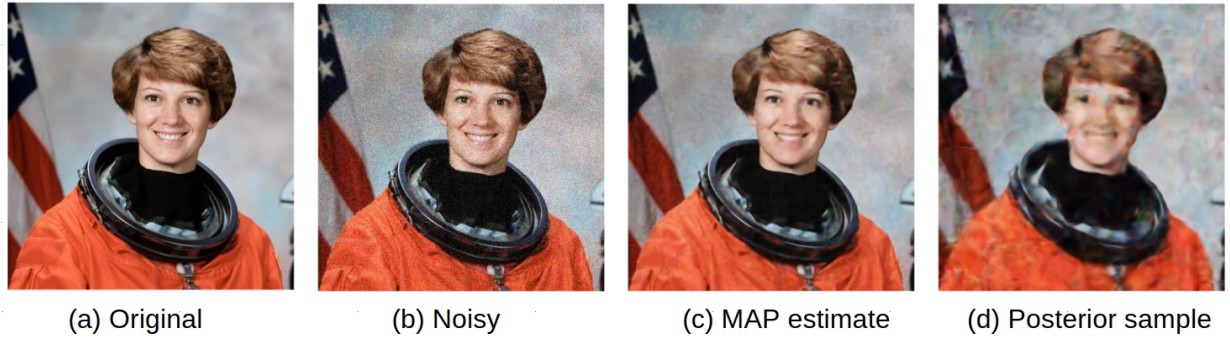

(a) Original      (b) Noisy      (c) MAP estimate      (d) Posterior sample

Figure 1: Illustration from Cagnotti (2023): A Deep decoder is trained to denoise the image shown in (b). (c) shows a MAP estimate obtained by training with Equation 4 using a Gaussian prior for weights, i.e., weight decay. (d) shows reconstruction with a sample drawn with MALA from the posterior distribution of the network weights after making sure the chain has converged. As explained below, this posterior sample likely comes from the typical set, and the reconstruction quality is clearly lower than the MAP estimate.

We find it important to explicitly define the "optimality" of these sets of weights. In this work, we adopt the following definition. If two sets of weights are optimal, they are both local minima of the loss function and yield similarly high accuracies on the training and/or validation sets. As such, it would not be possible to choose one over the other; thus, they are both optimal. Evaluating these different optimal models on the same sample would allow for approximating the epistemic uncertainty.

The current set of methods for estimating epistemic uncertainty can be divided into two groups. First, *the discrete support group*, estimates a finite number of parameter sets either through training multiple models and ensembling (Lakshminarayanan et al., 2017), approximating ensembling through Monte-Carlo (MC) Dropout (Gal & Ghahramani, 2016), or directly sampling from a posterior distribution using an appropriate sampling, such as Hamiltonian Monte Carlo (Betancourt, 2017). The second group, *the continuous support group*, estimates continuous posterior distributions for network parameters through using Laplace approximations (MacKay, 1992; Ritter et al., 2018; Daxberger et al., 2021) or variational inference (Graves, 2011; Bishop & Nasrabadi, 2006).

Setting model ensembling aside for a moment, the current techniques rely on an underlying Bayesian model and posterior distribution of network weights given a dataset. This posterior is either approximated or sampled from. However, even in small networks, the number of parameters is very high, and therefore, the underlying Bayesian model is very high-dimensional. As such, these Bayesian models are prone to counterintuitive effects of *typical sets* in high dimensions. An intuitive description of a typical set is given by Carpenter (2017); MacKay (2003) as *"...the central log density band into which almost all random draws from that distribution will fall"*. The main issue is that this band may be quite far from the modes of a distribution in high dimensions (Kirby, 1969). Thus, drawn samples will come from areas where the likelihood is potentially extremely low. When we consider the posterior distribution of network weights, this means sampled sets of weights may not be optimal as we defined above. They most likely have very low data likelihoods, or in other words, they will not make an accurate prediction on the training set. Cagnotti (2023) et al. demonstrated this on the problem of image denoising with the Deep Decoder model (Heckel & Hand, 2019). Denoising an image with network weights that are sampled from the corresponding posterior distribution, using a Metropolis-adjusted Langevin algorithm (MALA), yielded much worse results compared to the Maximum-a-Posteriori (MAP) estimate, which is illustrated in Figure 1.

Model ensembling by Lakshminarayanan et al. (2017) notably does not share this issue since the different sets of weights are all optimized and thus constitute modes in the posterior distribution when we view this approach from a Bayesian perspective using a flat prior (Wilson & Izmailov, 2020; Gustafsson et al., 2020; Pearce et al., 2018). While this is a good step, the small number of models that can be extracted with this approach may underestimate the true epistemic uncertainty, since there may be many more optimal weight sets than the number of models in the ensemble. One approach is to use Laplace approximations for each

model in the ensemble (Eschenhagen et al., 2021). Although the approach shows promising experimental results, the above-described problem of sampling in high dimensions persists here.

In this work, we propose a middle way between discrete and full continuous support by defining a mixture of *bounded support regions*. We propose to approximate the posterior locally around the weight configurations found by a deep ensemble using *locally bounded MALA* (lbMALA). We do so by defining a fixed-width hyperball around each ensemble weight parameter and setting a reflective boundary condition, effectively reducing the search space and support of the distribution to which the Markov Chain converges. While the normal distribution found by the standard Laplace approximation might struggle to pick up the complex statistical dependencies between weights, lbMALA inherently takes such dependencies into account.

To test whether lbMALA indeed leads to improved posterior estimates, we design a fine-grained OOD detection validation benchmark. This measures how sensitive different epistemic uncertainty methods are to samples that move gradually out of distribution. Finally, through the improved performance of lbMALA on this benchmark, we provide evidence that the typical set problem is indeed real and affecting the uncertainty estimates of state-of-the-art (SOTA) models.

## 2 Related Work

In probabilistic machine learning, the effects of concentration of measure and typicality appear both in the data and parameter space. The relevance is apparent: with larger models and higher-dimensional data, such as higher-resolution images, the studied spaces grow in dimension, reinforcing the issue. Nalisnick et al. (2018) and Choi et al. (2018) find that deep generative models might assign higher likelihoods to OOD data than to their training data. Based on this observation, Nalisnick et al. (2019) argues that high-likelihood samples might not be part of the typical set of the high-dimensional distribution of images. They developed an OOD test based on typical set membership, where the entropy of new samples is compared to the entropy of the source distribution using the condition in Equation 6. However, Zhang et al. (2021) finds the test unreliable, and Osada et al. (2024) attributes this to varying image complexity. Grathwohl et al. (2019) proposes a score that argues that data points with a high likelihood outside the typical set should have a higher gradient norm than ID samples. Recently, Abdi et al. (2024) has applied the above methods to a medical imaging setting with promising results in OOD detection.

The effects of typicality for Gaussian posterior approximations given by mean field variational inference in the parameter space of a neural network have been observed and discussed by Farquhar et al. (2020) and Farquhar (2022), which they termed "*soap bubble*" pathology. They propose an alternative probability distribution based on hyperspherical coordinates that forces probability mass to be close to the mean. However, such probability concentration may be "artificial" as samples are biased towards the mean with less exploration.

The combination of multiple MAP estimates from a deep ensemble with local approximations of the posterior around each estimate is a well-established technique. Eschenhagen et al. (2021) uses post-hoc Laplace approximations around independently trained neural networks, resulting in a Gaussian mixture model. The marginal likelihood is used to weight each distribution. Wilson & Izmailov (2020) extends Stochastic Weight Averaging (SWA) (Izmailov et al., 2018) and Stochastic Weight Averaging Gaussian (SWAG) (Maddox et al., 2019) to multiple neural networks of an ensemble. SWA averages the weights collected at different points, usually in later epochs, during training with a constant or cyclic learning rate. Wilson & Izmailov (2020) introduce MultiSWA, which uses an ensemble of models found with SWA. SWAG extends SWA by not only averaging the weights but also fitting a Gaussian distribution to them, capturing the posterior distribution locally. MultiSWAG from Wilson & Izmailov (2020) extends MultiSWA and involves training multiple neural networks independently and then applying SWAG to each of these networks.

The Bayesian supervised learning framework distinguishes two types of uncertainty: Aleatoric, which captures the inherent variation in the data, and Epistemic uncertainty, which informs about a trained model's state of knowledge in a certain area of the domain of the network. Different theoretically derived uncertainty measures capture aleatoric and epistemic uncertainty, and it is an open research question which of those measures captures the different types of uncertainty best (Schweighofer et al., 2023a; Wimmer et al., 2023; Zepf et al., 2024; Schweighofer et al., 2023b). The success of the chosen uncertainty measure can depend on

the application; therefore, mutual information, variance in predictions, the deviation of the posterior from the prior and predictive entropy (Abdar et al., 2021), could vary in their suitability downstream tasks like OOD detection. In addition, a strong correlation between measures of aleatoric and epistemic uncertainty has been found, posing the question of whether decomposition is possible (Kahl et al., 2024). Therefore, in practice, one often relies on the combined total uncertainty to circumvent the problem of decomposing uncertainties (Yang et al., 2024), especially when either the likelihood or the Shannon entropy of the predictive distribution is the most frequently used measure for total uncertainty in the Bayesian framework, which in practice is approximated by sampling using the Monte-Carlo method.

## 3 Method: Mixtures of locally bounded Langevin dynamics

### 3.1 Bayesian Deep Learning

The Bayesian framework for supervised learning assumes a data-generating process:

$$(x, y)_i \overset{\text{i.i.d.}}{\sim} p(x, y), \quad i = 1, \dots, N; \tag{1}$$

from which a dataset $D = (x, y)_i^N$ is an independently and identically distributed sample. To infer $y$ from $x$ we assume a model of $p(y|x)$ with parameters $\theta$ and search for likely model parameters $\theta$ based on the data $D$. The predictive distribution marginalizes over the model parameters

$$p(y|x, D) = \int p(y, \theta|x, D)d\theta = \int p(y|x, \theta)p(\theta|D)d\theta, \tag{2}$$

where the data $D$ and the model parameters $\theta$ are connected by Bayes' rule. The *posterior distribution* $p(\theta|D)$ of the parameters $\theta$ after observing the data $D$ then decomposes into

$$p(\theta|D) = \frac{p(D|\theta)p(\theta)}{p(D)}, \tag{3}$$

with $p(D|\theta)$ being the *likelihood* of observing the data $D$ given the parameters $\theta$, $p(\theta)$ the *prior distribution* of $\theta$, which represents the beliefs about $\theta$ before any data is seen and, the *evidence $p(D)$* is called marginal likelihood, that serves as a normalizing constant to ensure that the posterior distribution sums up to one. The integral $p(D) = \int p(D, \theta)d\theta = \int p(D|\theta)p(\theta)d\theta$ usually cannot be solved analytically and is intractable for numerical integration.

The standard way of training Neural Networks corresponds to MAP inference in the Bayesian framework and yields a single parameter configuration for the posterior distribution:

$$\theta_{\text{MAP}} = \arg_\theta \max \sum_{i=1}^{N} \log p(y_i|x_i, \theta) + \log p(\theta) \tag{4}$$

Here, $\log p(\theta)$ corresponds to a weight regularization, such as weight decay, and if no such regularization is used, it is a constant value corresponding to an improper flat prior. The goal of any approximate inference technique is to go beyond such point estimates to capture more characteristics of the posterior distribution.

### 3.2 Typicality

Consider a set of i.i.d. random variables $X_1, ..., X_n \overset{\text{i.i.d.}}{\sim} p(x)$. A sample of this set is a sequence of observed values $x^n = (x_1, ..., x_n)$. We call such a sequence typical (Cover, 1999) if it satisfies

$$2^{-n(H(x)+\epsilon)} \leq p(x_1, ..., x_n) \leq 2^{-n(H(x)-\epsilon)}, \tag{5}$$

for $\epsilon > 0$ and the Shannon entropy $H(x)$ of $p(x)$. The *typical set $A_\epsilon^{(n)}$* is defined as the set of all these typical sequences. If $x^n \in A_\epsilon^{(n)}$ we can rewrite the criterion in Equation 5 into:

$$H(x) - \epsilon \leq -\frac{1}{n} \sum_{i=1}^{n} \log_2 p(x_i) \leq H(x) + \epsilon, \tag{6}$$

where the Shannon entropy of a typical sequence is bounded from below and above by the Shannon entropy of the distribution $p(x)$ and defined by Equation 7.

$$H[p(y|x, D)] = -\sum_y p(y|x, D) \log p(y|x, D) \tag{7}$$

Since $-\frac{1}{n} \sum_{i=1}^{n} \log_2 p(x_i)$ converges to $H(x)$ in probability for large $n$ by the Asymptotic Equipartition property (Cover, 1999) it follows that:

$$P(x^n \in A_\epsilon^{(n)}) > 1 - \epsilon, \tag{8}$$

which means the probability that a sequence is part of the typical set is almost 1. At this point, it is pertinent to describe and emphasise the distinction between the regions of high probability (high density) and regions containing most of the probability mass, which is particularly important in high-dimensional settings. The probability density of a distribution indicates how likely an individual point is relative to others at the mode, but the probability mass depends on the density and volume of the region under consideration (Betancourt, 2017). To provide further intuition, Figure 2 illustrates a one-dimensional Gaussian distribution (on the left) divided into five regions based on its cumulative distribution function (CDF). Each colored region, therefore, has equal probability mass (0.2). If we draw 20 independent samples from this Gaussian, the probability that all samples fall within the central green region corresponding to the highest probability density is: $0.2^{20} \approx 1.4 \times 10^{-14}$. This probability rapidly decreases as the number of samples increases.

Conversely, the probability that at least one sample falls in either the blue or purple outer regions, which exceeds 0.9999, computed as: $1 - 0.6^{20}$ and increases with the number of samples. Interpreting these samples as coordinates of a 20-dimensional Gaussian vector, this implies that the probability that at least one dimension lies far from the high-density region is extremely high. Consequently, samples in high dimensions are far more likely to lie far from the Gaussian's mode than near it.

A second illustration (on the right in Figure 2) extends this idea by dividing the Gaussian into 25 equal-probability regions, each with probability 0.04. For a 20-dimensional Gaussian, the probability that at least one coordinate falls within the outermost blue or purple regions is: $1 - 0.92^{20} > 0.81$. For a 200-dimensional Gaussian, this probability exceeds 0.999999. Again, this demonstrates that in high dimensions, it is almost certain that at least one coordinate lies far from the mode, meaning typical samples occur far from the region of highest probability density.

Thus, in low dimensions $d$, Gaussian distributions have most of their probability mass close to the mean. However, for large $d$, an origin-centered normal Gaussian $\mathcal{N}(0, \sigma\mathbb{I})$ has almost all of its mass located near a thin annulus with radius $\sigma\sqrt{d}$ (Gaussian Annulus Theorem (Blum et al., 2020)). As a result, in high-dimensional distributions, most samples from a distribution will fall into this typical set (Carpenter, 2017). Considering the expected squared distance of a point $\mathbf{x} \sim \mathcal{N}(0, \mathbb{I})$ to the mean provides some intuition as to where to expect points when we sample from the distribution:

$$E(|\mathbf{x}|^2) = \sum_{i=1}^{d} E(x_i^2) = dE(x_1^2) = d \tag{9}$$

While the Annulus theorem, as a result of the concentration of measure, describes the probability mass, the entropy-based condition of the typical set inequation 6 leads to a similar result. For $n = 1$ a point $x$ belongs to the typical set of the distribution $\mathcal{N}(0, \sigma\mathbb{I})$ if $||x - \mu||_2^2 = \sigma\sqrt{d}$.

As a consequence, posterior approximations of Bayesian neural networks based on high-dimensional Gaussian's tend to under-represent the most-probable weight configurations (Farquhar et al., 2020; Bishop & Nasrabadi, 2006), since there is almost no probability mass in the hyper ball of radius smaller $\sqrt{d}$ and typical samples do almost certainly not lie in this region as illustrated in Figure 3. Assuming some degree of continuity in the weight space, i.e., weight configurations close to each other yield similarly performing

From One Dimension to High Dimensions in Gaussian Distributions

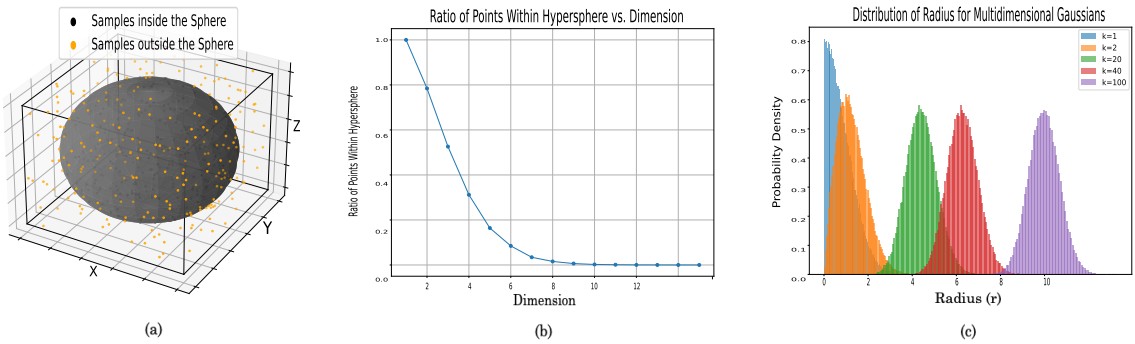

Figure 2: Illustration of high-dimensional Gaussian geometry. **Left:** A one-dimensional Gaussian partitioned into equal-probability regions using its CDF. **Right:** Extension to high dimensions, where multiple independent coordinates are sampled. As dimensionality increases, the probability that at least one coordinate lies in a low-density outer region approaches 1, indicating that high-dimensional samples concentrate far from the mode.

Figure 3: Illustrations of typical set "phenomena": (a) Samples of a uniform distribution on the unit cube in three dimensions. Samples that lie within the unit sphere are coloured grey, and samples outside are orange. (b) With growing dimensions of the unit cube, most samples lie outside of the respective unit sphere. (c) Probability density of a standard Gaussian distribution as a function of distance to the origin. In higher dimensions, most probability mass concentrates not at the mean (origin) but on a given radius.

functions, the likely weight configurations within the neighborhood of a MAP estimate would not be sampled by Gaussians in high dimensions.

## 3.3 Metropolis adjusted Langevin algorithm

The Metropolis-adjusted Langevin algorithm (MALA) is an MCMC method that combines Langevin dynamics (Welling & Teh, 2011) with the Metropolis-Hastings acceptance criterion to sample from a target distribution (Roberts & Rosenthal, 1998).

1. Langevin Proposal:

$$\theta' = \theta_t + \frac{\varepsilon^2}{2} \nabla_\theta \log \pi(\theta_t) + \varepsilon \eta, \qquad \eta \sim \mathcal{N}(0, I_d) \tag{10}$$

Where $\theta'$ is the proposed move, $\theta_t$ is the current position, $\nabla_\theta \log \pi(\theta_t)$ is the gradient of the log target distribution at $\theta_t$, $\varepsilon > 0$ is the step size, and $\eta$ is a $d$-dimensional standard normal random vector. We take $\pi(\theta) \propto p(y \mid \theta) \, p(\theta)$ to be the target posterior over the (flattened) parameter vector $\theta \in \mathbb{R}^d$.

The corresponding forward proposal density is

$$q(\theta_t \to \theta') = \mathcal{N}\left(\theta'; \ \theta_t + \tfrac{\varepsilon^2}{2}\nabla_\theta \log \pi(\theta_t), \ \varepsilon^2 I_d\right) \tag{11}$$

2. Metropolis-Hastings Acceptance Criterion:

$$\alpha(\theta_t, \theta') = \min\left(1, \frac{\pi(\theta') \, q(\theta' \to \theta_t)}{\pi(\theta_t) \, q(\theta_t \to \theta')}\right) \tag{12}$$

Where $\pi(\theta)$ is the target distribution and $q(\theta_t \to \theta')$ is the transition kernel of the Langevin dynamics. The term $q(\theta' \to \theta_t)$ represents the probability of transitioning from the proposed move $\theta'$ back to the current position $\theta_t$, and vice versa for $q(\theta_t \to \theta')$.

The algorithm accepts the proposed move $\theta'$ with probability $\alpha(\theta_t, \theta')$. If accepted, the chain moves to $\theta'$, otherwise it remains at $\theta_t$.

### 3.4 Domain Restriction through Reflective Boundary Condition

A hyperball is defined around the $\theta_{\mathrm{MAP}}$ with radius $R$, where $R < \sigma\sqrt{d}$ (Gaussian Annulus). Here, $\sigma$ is the standard deviation of a Laplace approximation, i.e., a function of the curvature in the $\theta_{\mathrm{MAP}}$. This curvature can also be calculated using the diagonal Hessian $H$ of the negative log-posterior (Bergamin et al., 2023). MALA is then run while enforcing a reflective boundary in this constrained region. Thus, if a proposed MALA step moves the chain outside the hyperball of radius $R$, the proposal is "reflected" back into the domain. This prevents the chain from leaving a feasible space. In this way, we still converge to a stationary distribution but within a restricted domain. This ensures more stable outcomes and confines the samples to regions with the highest probability, thereby ensuring an ergodic chain (Oliviero-Durmus & Moulines, 2024). A key consideration in this method is that one can not simply choose an $\varepsilon$ that scales proportionally to $\sqrt{d}$ as choosing a smaller $\varepsilon$ may seem to address the issue at first; however, unfortunately, it does not. Decreasing $\varepsilon$ will yield samples from the Markov Chain that are too similar to each other. In other words, the autocorrelation of samples in the Markov Chain increases as $\varepsilon$ decreases. The samples will not cover the posterior and thus will lead to an underestimation of the actual variability, in the context of this work, the epistemic uncertainty. Here, the approach restricts the posterior distribution but still allows the model to move with longer steps within that area to select samples. The selected samples, therefore, are less similar to each other compared to simply using a smaller $\varepsilon$.

### 3.5 Mixtures of Locally Bounded Langevin dynamics

The approach starts with an ensemble of initial models, represented from a constrained region of high probability. Each model is run under a domain-restricted MALA, yielding a locally bounded stationary distribution in the region. This results in a mixture of the distributions (lbMALA) in multiple high-probability regions.

### 3.6 Implementation

**lbMALA was implemented** in three stages, as illustrated in Algorithm 1:

**Step 1: Initialisation** A pre-trained baseline model is required and used to initialise the chain with a starting point estimate, $\theta_0$, i.e., the MAP estimate. The associated model parameters are loaded and

hyperparameters defined; step size $\varepsilon$, chain length $L$, burn-in $B_{\text{burn}}$. Then local curvature was estimated using the Hessian of the negative log posterior, to define the radius of the reflective hyperball as $R = \beta \left\| \sigma \right\|_2$, radius scaling factor $\beta = 2$ (two standard deviations), with centre $c = \theta_0$, to dynamically calculate the reflective boundaries and constrain the sampling to a high probability region of the posterior, described in Algorthim 1. A buffer size $S = 5000$ was also initialised to store a sufficient number of samples while searching within the boundary conditions.

**Step 2: Constrained MALA Sampling with Reflective Boundaries** aimed to accelerate convergence, but through a constrained MALA adapted with the reflective boundary condition to provide an efficient sampling approach. With each iteration, a mini-batch of data, $\mathcal{D}$ is drawn, the gradient $g = \nabla_\theta \log \pi(\theta)$ is computed, and a new Langevin sample is proposed $(\theta')$. Each proposal is checked against the hyperball constraint, and if it lies outside the ball, it is radially reflected back toward the MAP centre, as detailed in Section 3.3. This ensures better definition in the constrained region. Thereafter, the continuous acceptance and rejection rates are monitored and assessed as convergence occurs to ensure stability and convergence in the bounded region.

**Step 3: Extract Posterior Samples and Fine-Tune** allows $n = 5$ posterior samples from the chain, with the sample initialising a separate model which was fine-tuned and trained for a further 10 epochs for MNIST and 100 epochs for CIFAR-10. To ensure consistency with SWAG and Multi-SWAG, the baseline MAP at 10 epochs was utilised for MNIST and 100 epochs for CIFAR-10. A final set of parameters produced the optimal lbMALA approach with step size $\varepsilon$ of $1e-6$ for MNIST and $1e-8$ for CIFAR-10, chain length $L$ of 400, burn-in $B_{\text{burn}}$ of 1000 samples, with the learning rate for the final optimisation set to $lr = 1e-4$. Given the computational cost of a full hyperparameter search, we limited optimisation to adjusting the learning rate between $lr = 1e-4$ and $lr = 1e-6$ for MNIST and $lr = 1e-4$ and $lr = 1e-8$ for CIFAR-10, with both datasets having a burn-in of 1000 to 2500 samples. Interestingly, empirically, we observe that acceptance rates were initially low when the chain was far from the typical set. Still, we observed that it stabilises during burn-in and sampling, with rejection becoming comparatively infrequent. We illustrate this transient-to-stable behavior with a representative run shown in Figure 8, Appendix A.

## 4 Experimental Analysis

Our experiments are designed to highlight the ability of different methods to perform fine-grained ID versus OOD detection. To this end, we utilise two datasets, the Morpho-MNIST toolkit was used to create versions of the MNIST digit classification dataset that are increasingly distorted (Castro et al., 2019). Second, corrupted CIFAR-10 (CIFAR-10-C) was used to obtain increased corruption severity on the CIFAR-10 dataset (Hendrycks & Dietterich, 2019). Using these increasingly OOD datasets, we quantify the ability of different posteriors to rank these datasets according to their level of OOD distortion.

---

**Algorithm 1** lbMALA Algorithm

---

1: **Step 1: Initialisation**
2: Load pretrained model with MAP estimate $Z_0$, flattened to $\theta_0$
3: Define hyperparameters: step size $\varepsilon$, total iterations $L + B_{\text{burn}}$, buffer size $S$ and radius scaling factor $\beta = 2$ (two standard deviations).
4: Estimate local curvature using the diagonal of the Hessian $H$
5: Compute approximate marginal scales

$$\sigma_i = \frac{1}{\sqrt{H_{ii}}}, \quad i = 1, \ldots, d$$

and collect as vector $\sigma = (\sigma_1, \ldots, \sigma_d)$ in order to approximate local posterior variability
6: Define hyperball center and radius:

$$c \leftarrow \theta_0, \qquad R \leftarrow \beta \, \|\sigma\|_2,$$

where the norm

$$\|\sigma\|_2 = \sqrt{\sum_{i=1}^{d} \sigma_i^2} \approx \sigma \sqrt{d}$$

corresponds to the Gaussian-annulus radius in $d$-dimensional parameter space, where $R$ is the fixed radius of the locally bounded region, and the feasible set is

$$\mathcal{B} = \{\theta : \|\theta - c\|_2 \leq R\}$$

7: Initialize $\theta \leftarrow \theta_0$, rejection counter $\leftarrow 0$, buffer $\mathcal{S}$
8: **Step 2: Constrained MALA Sampling with Reflective Boundaries**
9: **for** $i = 1$ to $L + B_{\text{burn}}$ **do**
10:     Draw mini-batch $(x, y) \sim \mathcal{D}$
11:     Compute gradient of posterior for batch:

$$g = \nabla_\theta \log \pi(\theta)$$

12:     Propose a new state (MALA proposal):

$$\theta' = \theta + \frac{\varepsilon^2}{2} g + \varepsilon \, \eta, \qquad \eta \sim \mathcal{N}(0, I)$$

13:     Apply reflective boundary:
14:     **if** $\|\theta' - c\|_2 > R$ **then**
15:         Project $\theta'$ back onto the boundary of the hyperball (approximating radial reflection)

$$\theta' \leftarrow c + R \frac{\theta' - c}{\|\theta' - c\|_2}$$

16:     **end if**
17:     Compute gradient at the proposal:

$$g' = \nabla_\theta \log \pi(\theta')$$

18:     Compute acceptance ratio using proposal densities (using Eq. 11) :

$$\alpha = \min\left(1, \frac{\pi(\theta') \, q(\theta' \to \theta)}{\pi(\theta) \, q(\theta \to \theta')}\right)$$

19:     If it is accepted, set $\theta \leftarrow \theta'$, else keep previous $\theta$.
20:     **if** $i > B_{\text{burn}}$ **then**
21:         Append $\theta$ to buffer $\mathcal{S}$.
22:     **end if**
23: **end for**
24: **Step 3: Extract Posterior Samples and Fine-Tune**
25: Select $n = 5$ posterior samples from $\mathcal{S}$; for each and initialise a model
26: Retrain model with $\mathcal{S}_s$ for additional epochs
27: **Output:** Ensemble of retrained models

---

### 4.1 Design and Setup

The Morpho-MNIST toolkit provides four different distortions that can be applied to the original MNIST digits to produce either a thickened, swelled, thinned, or fractured (broken continuity) version of any MNIST digit, see Figure 4 for an illustration. As shown in the figures, the level of distortion is controlled by a perturbation factor that influences the severity of a distortion. We used the original digits as an ID (Plain) dataset for training and ID testing, and generated, for each of the four distortions, three increasingly OOD test sets by using each distortion with three increasing severity levels.

The CIFAR-10-C dataset consists of images derived from the original CIFAR-10 test set by applying a diverse set of image corruptions. In total, 19 corruption types are considered, with categories such as additive noise, blur, weather effects, and digital distortions. Each corruption applies five increasing severity levels to control the strength of the perturbation. Thus, a comprehensive benchmark of distributional shifts that systematically degrade image quality is developed. In our experiments, the original CIFAR-10 test set is used as the ID evaluation set, while the corrupted variants at increasing severity levels serve as OOD test sets for assessing robustness under progressively more challenging severities. Overall, there are ten object classes. We utilise one corruption from each of the five different categories: impulse noise, gaussian blur, snow, elastic transform, and spatter for all five levels of severity, illustrated in Figure 5.

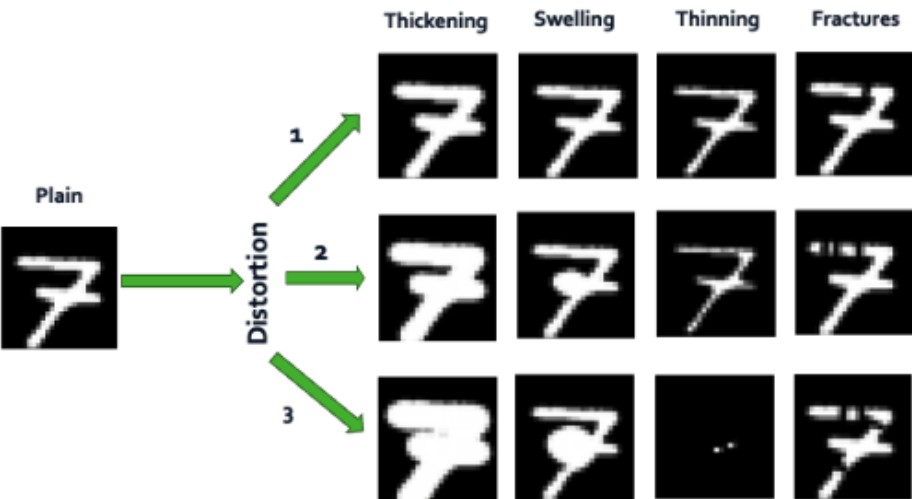

Figure 4: An illustration for the digit 7 across the types of perturbations, across each distortion placed on the original "*plain*" MNIST dataset.

To evaluate the performance of lbMALA, it was benchmarked against several baseline methods for epistemic uncertainty quantification. All methods were tested with two different backbones: a lightweight convolutional neural network (CNN) and a more complex ResNet18 model. The CNN model consists of 2 convolutional layers with ReLU + max pooling, followed by 2 fully connected layers producing logits for the 10 digits (classes). The ResNet18 model comprises 18 layers, 4 residual blocks, global pooling, and a final fully connected layer. Both models were trained on the original MNIST (Plain) dataset, consisting of $28 \times 28$ grayscale images, of which 60.000 examples were used for training (90%) and validation (10%), and 10.000 were used for testing. The same hyperparameters $lr$ and $b$ were used for both the CNN and ResNet architectures. For CIFAR-10, we employed the same model architectures, modified to accommodate $32 \times 32$ RGB images. We used the same training–validation split as for MNIST, with 45.000 images for training and 5.000 for validation. For evaluation, 10.000 test images were used for each corruption severity within each category. In addition, we implemented an unbounded MALA using identical parameter settings to enable a direct comparison with lbMALA.

Figure 5: An illustration for the class horse to indicate the five categories of corruptions and levels of severity.

**Baselines**  To evaluate the performance of the **lbMALA** method for Bayesian posterior sampling and uncertainty quantification, we performed comparisons against a range of widely utilised baselines. All baselines were selected to cover a range of methods, including deterministic point-estimate models and Bayesian inference approaches. Each method was implemented under consistent training conditions to ensure a robust and fair comparison.

**Maximum A Posteriori (MAP)** was used as the main baseline. Here, the cross-entropy loss is optimised, and a single point-based estimate of model weights is generated as a result of the corresponding zero-mean Gaussian prior over model parameters. This provides a computationally efficient approach, but as mentioned earlier provides overconfident outcomes as there is no measure of uncertainty. Thus, it is used as an initial starting point for subsequent approaches.

The **MC dropout**, was implemented to compute an approximation of variational inference by randomly dropping units in the model to prevent overfitting (Gal & Ghahramani, 2016), (with a dropout probability of $p$=0.3) for both MNIST and CIFAR-10. However, even though it was easy to implement, it has been known to not provide the best representation of model uncertainty.

**Deep Ensembles,** utilises multiple independently trained baseline MAP models (Lakshminarayanan et al., 2017) , while computationally more expensive, typically outperform MC dropout, whose weight configurations tend to be less diverse (Durasov et al., 2021). We trained 5 models, each initialized and trained from scratch on the same training data. All three of these approaches were trained over 20 epochs and with a learning rate of $lr = 1e - 3$ using stochastic gradient descent (SGD) and a batch size of 64 for MNIST. CIFAR-10 utilised a batch size of 128, an SGD learning rate of $lr = 1e - 1$ over 200 epochs. All

subsequent models trained and discussed utilised the same batch size of 64 and 128, for the MNIST and CIFAR-10 models, respectively.

**SWAG** provides a structured and more rigorous approach to Bayesian approximation. Here, SWAG computes multiple weights when training using SGD and uses the mean and covariance of the collected weights to estimate a Gaussian distribution; samples from this distribution are used to provide scalable yet high-quality uncertainty estimates (Maddox et al., 2019). We used the baseline MAP model at 10 epochs as the initial starting MAP point for MNIST and 100 epochs for CIFAR-10. However, even with a better approximation, SWAG is centered around the MAP estimate and thus may still not capture model diversity. We therefore extended the approach through **Multi-SWAG**, which utilises an ensemble of SWAG model to improve posterior coverage and improve prediction uncertainty with a more diverse posterior (Wilson & Izmailov, 2020). Both SWAG and Multi-SWAG utilised a learning rate of $lr = 1e-2$ for MNIST and CIFAR-10.

The last comparative approach implemented was the **mixture of Laplace approximations** (Eschenhagen et al., 2021), applied to the baseline of ensemble MAP models to represent the complexity of the posterior distribution using multiple local optima in the loss landscape. This approach allows a more accurate quantification of uncertainty using the multimodal nature of the posterior. A smaller noise perturbation was used (0.01) on model parameters, as higher values created very noisy and destabilised predictions for both MNIST and CIFAR-10.

### 4.2 Results

To investigate the relative performance of lbMALA against the Bayesian SOTA approaches for fine-grained OOD detection, the entropy values, reflecting epistemic uncertainty, were used to classify samples as ID or OOD. For a well-estimated posterior distribution, we would expect our ability to detect distorted samples of either kind as being OOD to increase with increasing level of distortion or severity for MNIST and CIFAR-10, respectively. We illustrate our performance on ID vs OOD performance, quantified using the Area Under the Curve (AUC) metric, for various distortions in Figure 6, with CNN results in the top row and ResNet18 on the bottom row for MNIST and for various corruptions and their severity in Figure 7 for CIFAR-10. Finally, all log entropy plots over the ID and OOD test sets across all distortions are represented in Appendix A for the MNIST dataset.

We note that while all the tested methods show the desired trend of increased AUC for ID vs OOD classification as the distortion level increases, our lbMALA performs consistently well across both distortion types and backbones on both the MNIST and CIFAR-10 datasets.

## 5 Discussion

Quantification of epistemic uncertainty is critical in many applications, such as the OOD detection problem tackled here. Many Bayesian approaches rely on posterior approximations based on posterior samples, and are therefore sensitive to the fact that typical sets of very high-dimensional distributions can fall far from the posterior modes. As a result, these samples do not represent sets of optimal parameters for the task at hand. Our proposed lbMALA approach, which utilizes sampling within a restricted space around MAP estimates, will generate samples closer to the MAP, which are more likely to be optimal for the ID datasets. These samples should yield stronger differences between ID and OOD samples. The results presented in Figure 6 and Figure 7 confirm this assumption, as lbMALA has a more consistent and reliable performance across both backbones. In particular, the Figures suggest that entropy-based measures assist in identifying OOD cases of increasing complexity, where the distortions are increasingly pronounced, but the digit itself is still recognisable.

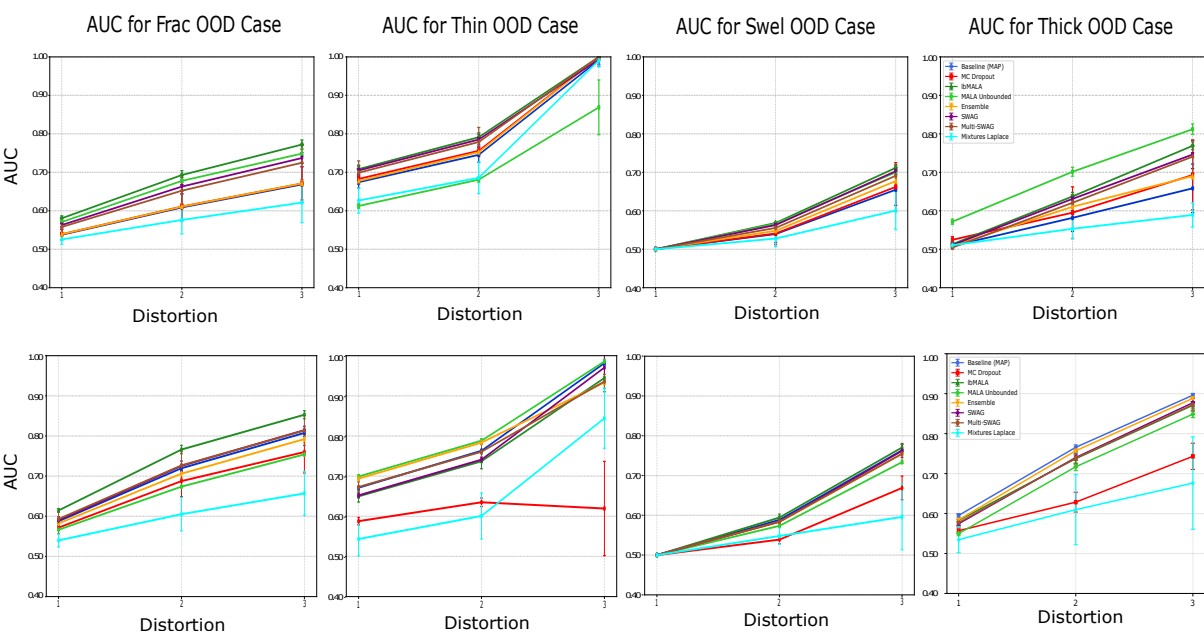

Figure 6: The AUC performance for MNIST with the CNN (top) and ResNet18 (bottom) architectures across all methods for fine-grained ID and OOD separation for (from left to right) Fractures, Thinning, Swelling, and Thickening datasets for 5 random seed initialisations for reliability.

**Differences to existing methods:**

**MNIST:** We compared lbMALA to a range of popular and recent methods for epistemic uncertainty quantification on our entropy-based fine-grained ID vs OOD detection task, including an Unbounded variant of MALA. MC dropout and Ensembles are well-known and widely used methods for epistemic uncertainty quantification. Despite providing only discrete support, they are both able to separate ID and OOD data in a fine-grained manner. However, their performance is not consistent between the 3 different distortions, as illustrated in Figure 6. For instance, while MC Dropout for the CNN architecture performed well and closer to other methods, it comes out as consistently worst on all distortions using the ResNet18 backbone, see Figure 6.

For the Thinning dataset in Figure 6, the MC dropout model exhibits a sharper deterioration in AUC performance. This may indicate that the ResNet18 architecture assigns high softmax scores to an incorrect class, leading to a reduced entropy value.

For the CNN backbone, lbMALA came out consistently top or second for the Fracturing, Thinning, and Swelling distortions. It gets more competition for the ResNet18 backbone, where SWAG and Ensemble also perform well. Interestingly, while lbMALA demonstrates consistently strong performance, the other methods vary more in their performance across backbones – in particular, as we expect the (fine-tuned) ResNet18 to represent a more over-fitted model than the simpler CNN.

To further analyse the performance of lbMALA across both architectures, we implemented a weighted ranking utilising the mean AUC's across distortions and dataset perturbations. Distortion 1 contributes a larger share of the weighting formula, ensuring that fine-grained OOD AUC performance has a larger influence on identifying the better-performing model. The final result is presented in Figures 11 and 12 in Appendix A, illustrating that under this weighting scheme, lbMALA demonstrates consistent and stable performance across architectures.

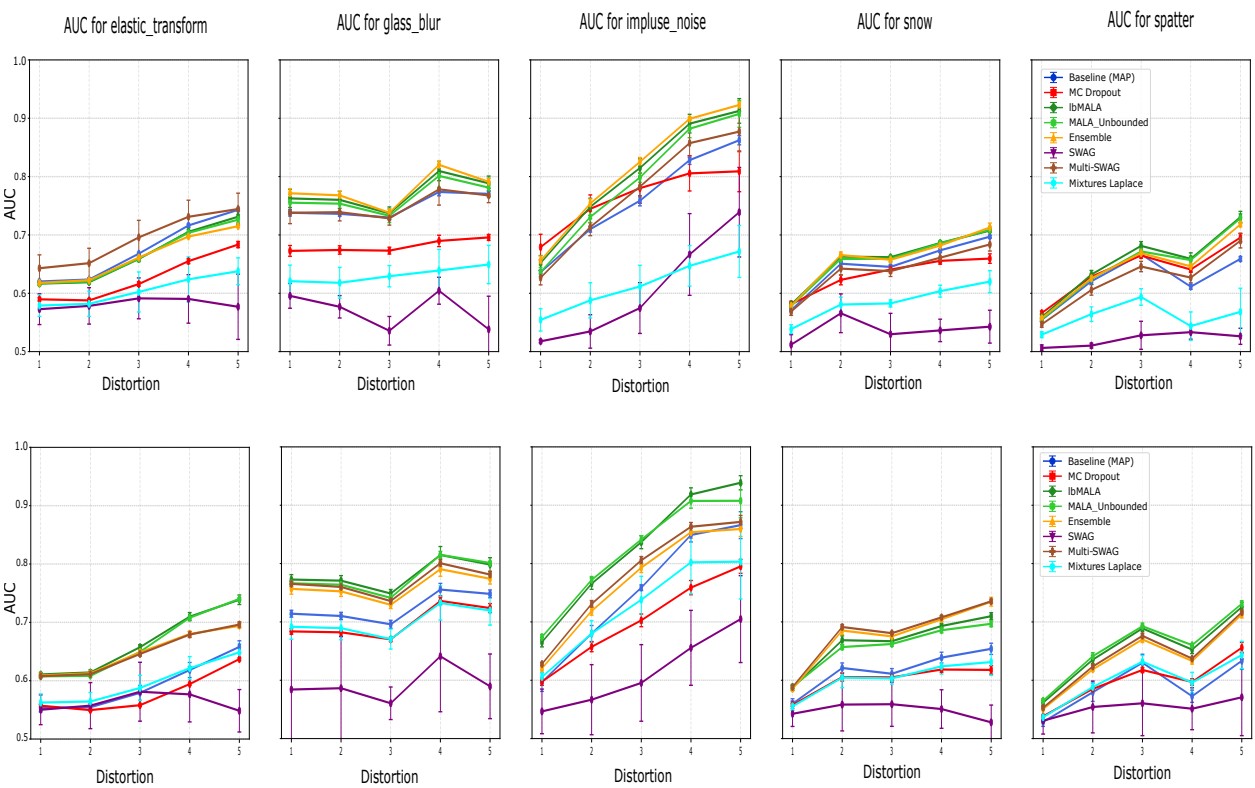

Figure 7: The AUC performance for CIFAR-10 for the CNN (top) and ResNet18 (bottom) architectures across all methods for fine-grained ID and OOD separation for (from left to right) five levels of severity for five corruptions. For each test, we ran 5 random seed initialisations for reliability.

Lastly, to evaluate and assess the model's behaviour under a far-OOD task, we evaluated the models using the Fashion MNIST dataset. The results for both CNN and ResNet architectures indicate that the ResNet backbone performs the best at OOD separation with lbMALA consistently indicating its ability to generalise well even in far-OOD settings, as illustrated in Figures 13 and 14, Appendix A.

**CIFAR-10:** Similar to the MNIST experiments, we compare against recent methods for epistemic uncertainty quantification in fine-grained ID versus OOD detection. Figure 7 shows that for the CNN backbone (top row), lbMALA consistently ranks among the top-performing methods, but, with stronger competition from ensemble-based approaches for the glass blur and impulse noise corruptions.

When comparing CNN and ResNet backbones, we observe a systematic increase in absolute AUC across all corruptions as distortion severity increases. For the ResNet backbone in particular, lbMALA exhibits lower variance across all five seeds and remains an overall top-performing method, similar to the behaviour observed on MNIST. This trend highlights the robustness of lbMALA, demonstrating strong sensitivity to increasing corruption severity, especially for noise-based and geometric distortions. While ensemble and MSWAG methods offer competitive performance for the CNN backbone, lbMALA emerges as a consistently reliable OOD detector across architectures.

**Limitations.** lbMALA does exhibit some limitations, such as the high computational costs for the two-step gradient evaluation (first proposing and then accepting a sample). While this is manageable for simpler problems like digit classification or lightweight architectures, it could impact more complex problems. However, as shown in Figures 15 and 16, Appendix A, the evaluation time for lbMALA is comparable to other competing methods, but importantly, lbMALA delivers the overall more consistent performance across OOD detection and architecture. Thus, we learn from lbMALA's positive performance, **that the typicality problem is real and affects the performance of epistemic uncertainty quantification**. This is critical knowledge

and a necessary starting point for developing more efficient methods whose solutions are not limited to the typical set.

**Need for fine-grained OOD detection.** In this paper, we have introduced a notion of fine-grained ID vs OOD detection, including a validation benchmark based on the Morpho-MNIST toolkit and CIFAR-10 dataset. This is a contrast to the standard validation schemes found in uncertainty quantification papers, where the performance of OOD detection is often demonstrated by showing that methods can recognize that data comes from an entirely different dataset – e.g., MNIST versus Fashion-MNIST. We stress that fine-grained OOD detection is of crucial importance, e.g. in healthcare AI where "typical" model failure does not come as a clear breakdown on obviously incorrect data – but rather as somewhat reduced performance on underrepresented population groups or disease subtypes (Cui & Wang, 2022). Often, these somewhat reduced performances are only visible when aggregating performances across an entire group, which is rarely done in everyday practice. As a result, fine-grained OOD detection is important for warning users of a potential decrease in the reliability of the model.

## 6 Conclusion

Ensuring more robust discrimination between ID and OOD data is fundamental for developing reliable deep learning models, more so for applications in high-risk settings where misclassification of samples may have a pronounced impact.

In this paper, we propose lbMALA, a Bayesian approach utilising a reflective boundary condition to enhance the ability to localise the approximation of the posterior weight configurations. To evaluate the performance of the method, various comparative SOTA approaches were implemented on the problem of fine-grained ID and OOD detection settings. Our analysis demonstrated promising outcomes with the novelty of lbMALA outperforming SOTA in terms of reliability and consistency across OOD datasets to separate data on a finer scale, as reflected by the AUC values through logistic regression analysis.

However, even with the noticeable improvements, some limitations require further exploration to determine the impact on computational costs and performance on more complex datasets. Nevertheless, it would be valuable to apply the lbMALA approach to a larger dataset or a more complex domain (medical applications) to explore further and evaluate its benefit for reliable uncertainty quantification.

## 7 Acknowledgments

We would like to acknowledge Roberto Cagnotti for his contributions to this line of research. His work served as an important source of inspiration for the ideas developed in this paper.

## 8 Impact statement

This paper presents work whose goal is to advance the field of Machine Learning. There are many potential societal consequences of our work, one of which we would like to emphasise. If the uncertainty estimates of our predictions are inaccurate and users trust them, it could mislead users and impact their work.

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

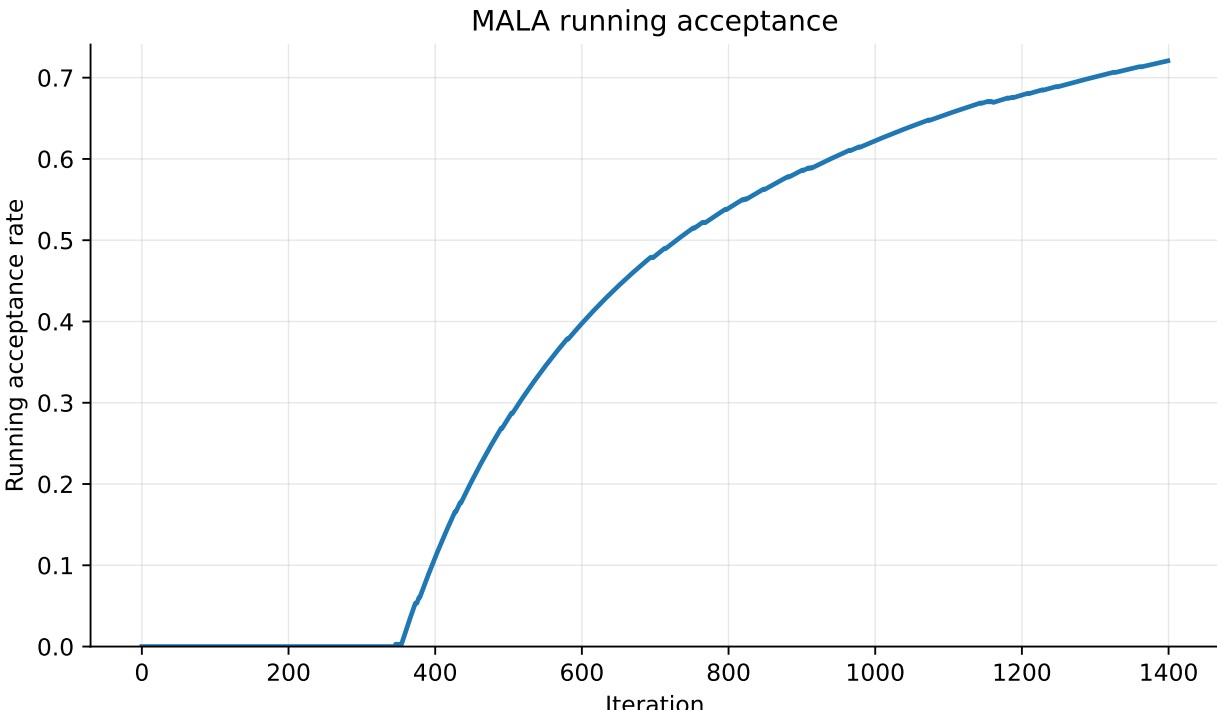

Figure 8: Illustrating the running (cumulative) acceptance rate of the MALA sampler over iterations.

## A    Appendix

In this appendix, we provide all the detailed log softmax entropy graphs for each of the methods discussed in this paper.

### A.1    Illustrations ID vs OOD

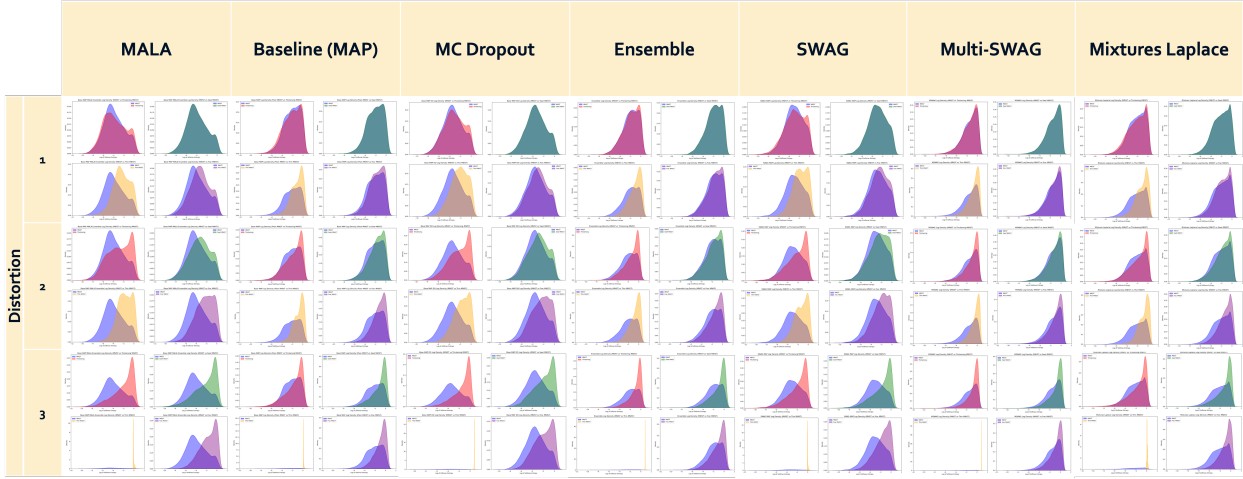

Figure 9: All log softmax entropy distributions indicating the ID and OOD datasets (Thickening, Swelling, Thinning, and Fracture datasets respectively) across all implemented methods for the CNN architecture.

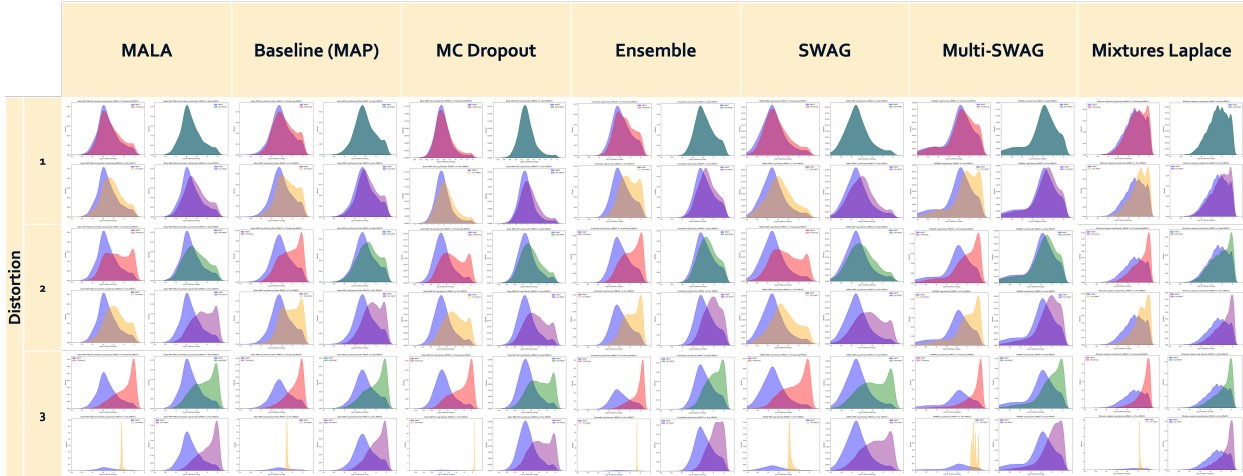

Figure 10: All log softmax entropy distributions indicating the ID and OOD datasets (Thickening, Swelling, Thinning, and Fracture datasets respectively) across all implemented methods for the ResNet18 architecture.

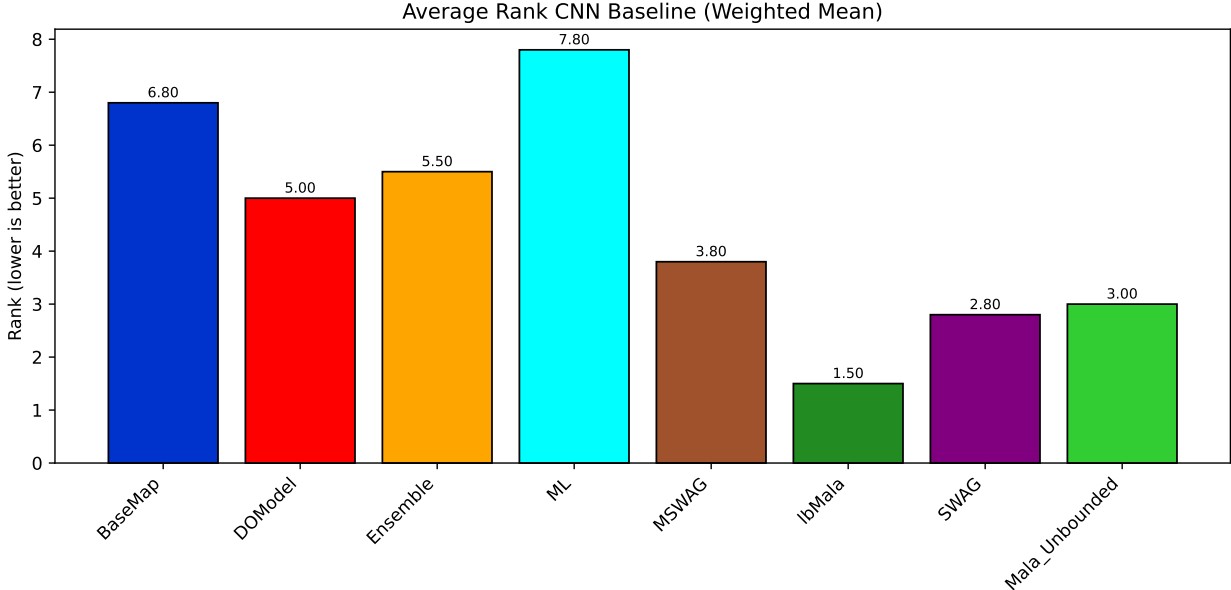

Figure 11: Average weighted AUC mean ranking of all uncertainty estimation methods on the CNN baseline architecture.Each bar represents the method's overall ranking across fine-grained OOD distortion types, with lower values indicating superior performance.

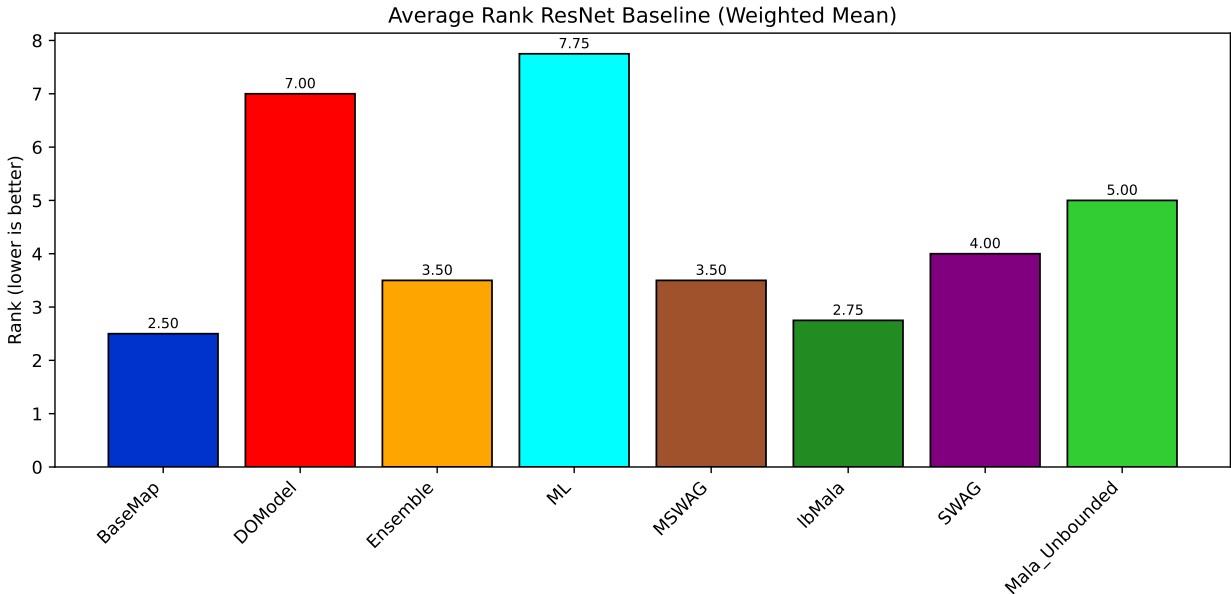

Figure 12: Average weighted AUC mean ranking of all uncertainty estimation methods on the ResNet baseline architecture.Each bar represents the method's overall ranking across fine-grained OOD distortion types, with lower values indicating superior performance.

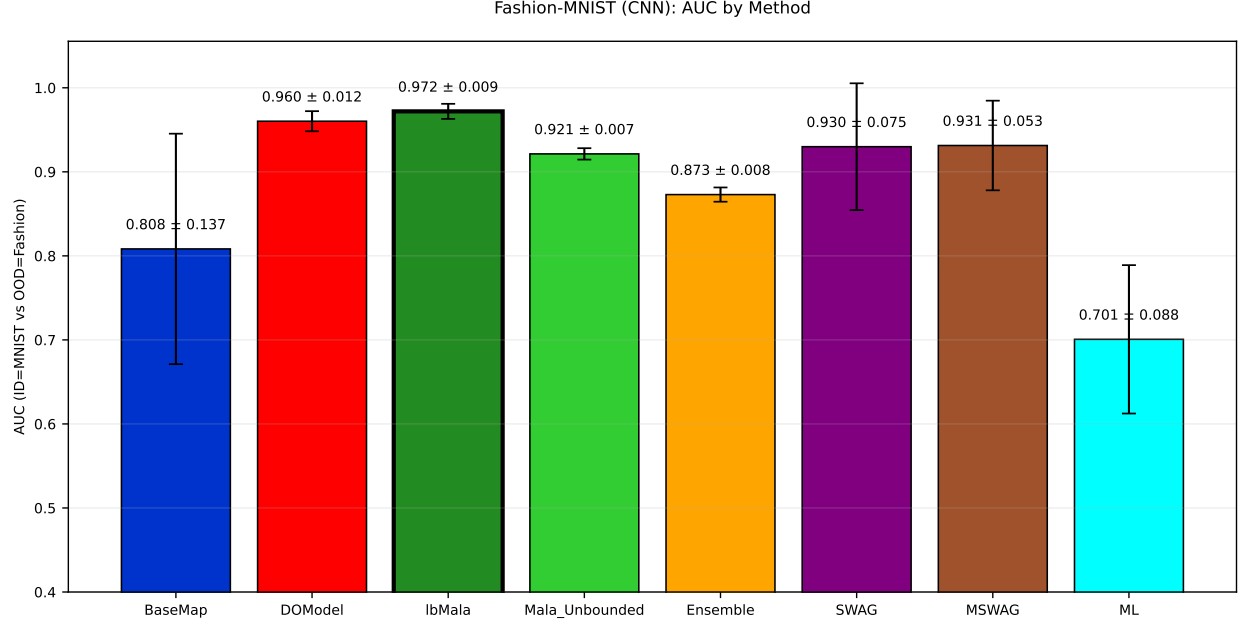

Figure 13: Far-OOD performance on the Fashion-MNIST dataset for all methods for CNN Baseline. Higher AUC values indicate better separation between MNIST (ID) and Fashion-MNIST (OOD).

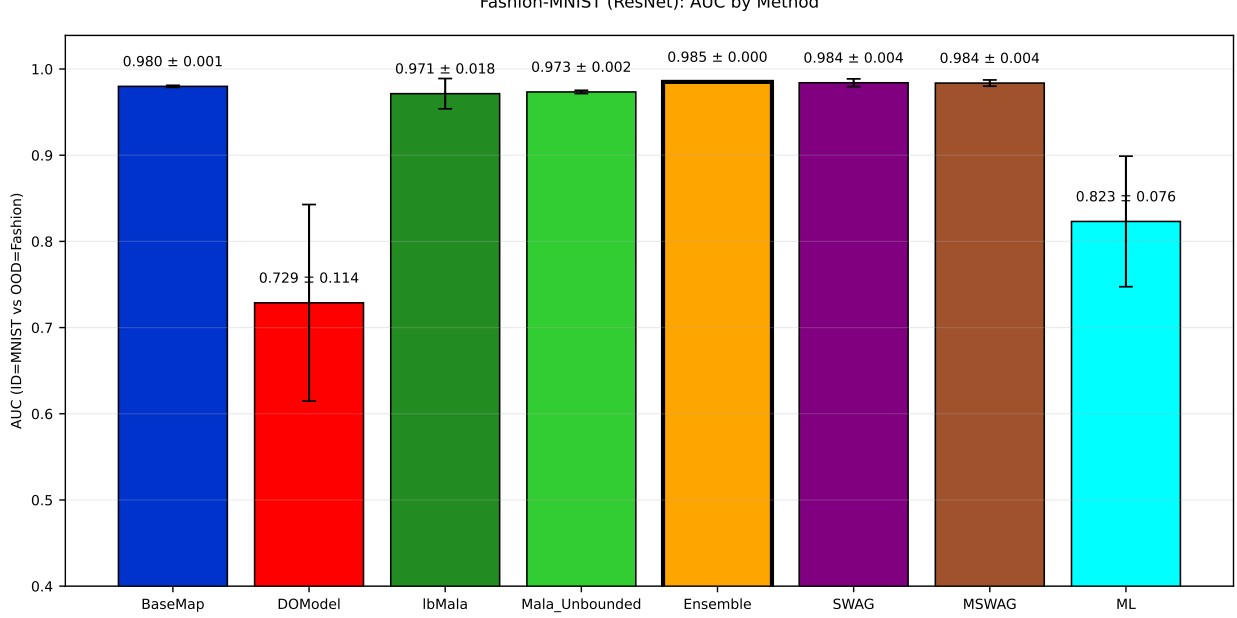

Figure 14: Far-OOD performance on the Fashion-MNIST dataset for all methods for ResNet Baseline. Higher AUC values indicate better separation between MNIST (ID) and Fashion-MNIST (OOD).

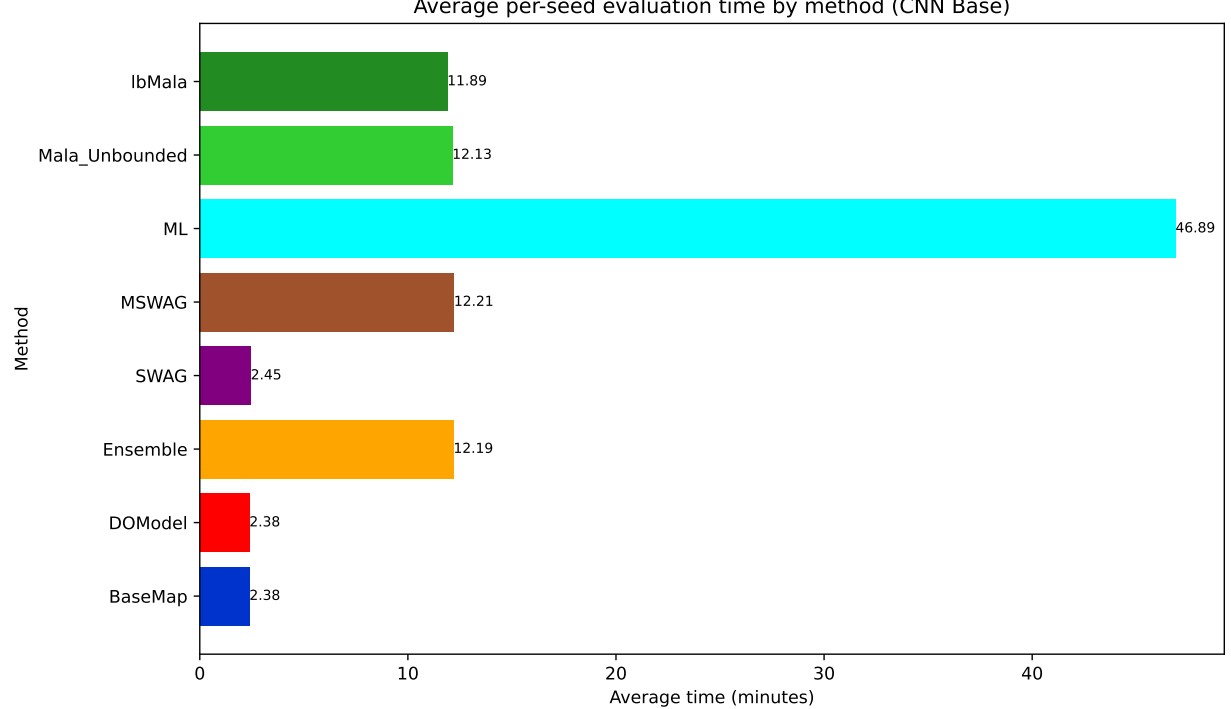

Figure 15: Comparison of evaluation times across methods, showing the average minutes required for the CNN architecture.

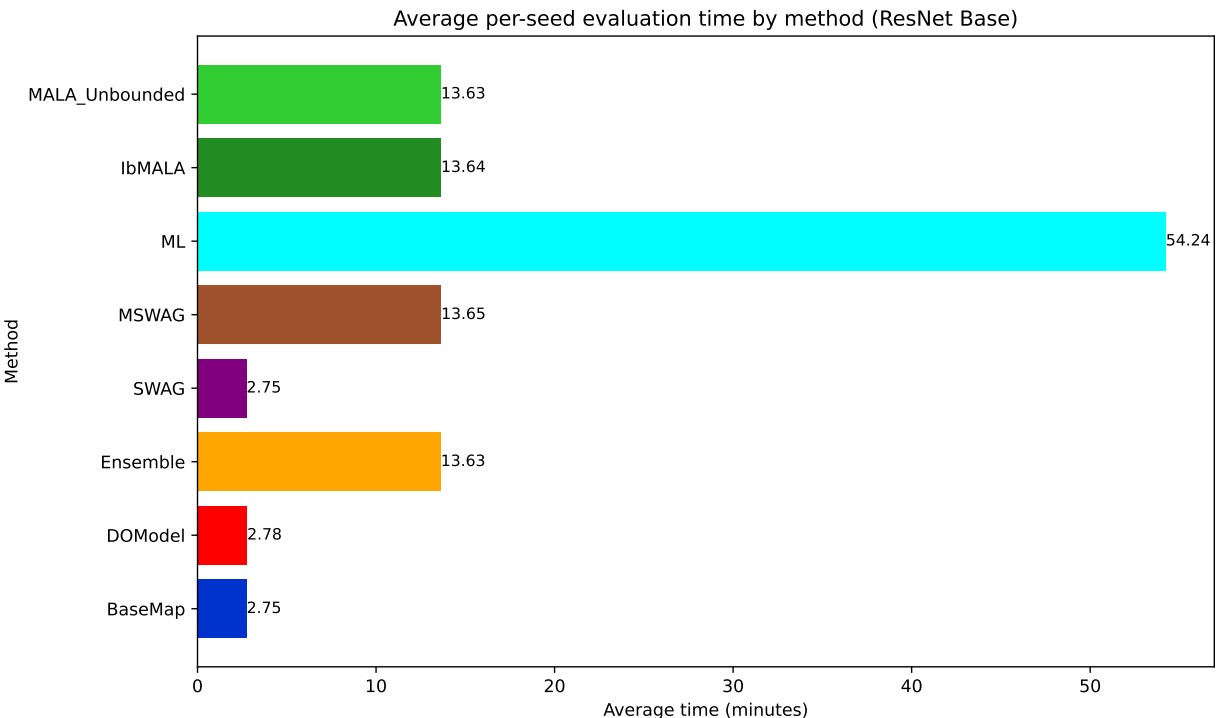

Figure 16: Comparison of evaluation times across methods, showing the average minutes required for the ResNet architecture.

