# OpenReview forum: "Mixtures of Locally Bounded Langevin dynamics for Bayesian Model Averaging"
_TMLR — Accepted by TMLR_

### Review · Reviewer_pGnx · 2025-10-27

**Summary Of Contributions:**

This paper introduces a locally bounded Metropolis-adjusted Langevin algorithm (lbMALA) for posterior approximation of parameters in Bayesian machine learning. The method restricts MALA proposals to remain within a predefined hypersphere around the MAP estimate, reflecting any proposals that fall outside back into said hypersphere. The motivation is that, for high-dimensional Gaussian distributions, samples are typically far away from the mean (as motivated in the paper with typical sets), which could lead to infeasible parameter configurations.

Strengths:
* Provides clear intuition about why sampling from high dimensional Gaussians could be far away from the mean/MAP estimate of the parameters via typical sets.
* The method is straightforward to implement, as it is a bounded modification of a well-established MCMC algorithm.

Weaknesses:
* I find the methodological contribution to be relatively incremental, consisting mainly of bounding MALA.
* A direct comparison with unbounded MALA is missing, which makes it less clear whether the bounding procedure improves the method.
* The experimental evaluation is limited, consisting of only one dataset (Morpho-MNIST).

**Additional Comments:**

In summary, the idea of constraining Gaussian samples within a hypersphere in parameter space is interesting. However, I find lbMALA represents a relatively incremental methodological contribution, and a broader range of experiments would be needed to convincingly demonstrate that the proposed bounding procedure leads to meaningful improvements.

**Audience:**

Yes

**Audience Explanation:**

The discussion of typical set behavior will be of interest to the Bayesian ML community. However, the practical contribution is modest, since lbMALA is a bounded variant of MALA. The overall impact is therefore limited but I believe a discussion on using such bounded methods could be interesting.

**Broader Impact Concerns:**

I don't have any concerns on broader impact of this work.

**Claims And Evidence:**

No

**Claims Explanation:**

Arguably, the main claim of the paper is that for high dimensional Gaussians, samples lie far away from the mean, therefore in Bayesian neural networks where the parameter space is high dimensional, this leads to infeasible proposals for parameters.

This claim is well motivated and clearly explained through the concept of typical sets. However, the claim that the empirical performance of lbMALA provides evidence of the typical set problem is less convincing. The method is not compared to standard (unbounded) MALA, so it remains unclear whether the observed improvements stem from the bounding procedure itself. Moreover, if bounding improves performance, one might expect similar gains for other Gaussian-based methods, which is not investigated.

Since the primary contribution of the paper is algorithmic, I would expect more extensive experimentation to substantiate the claims. Currently, only one experiment is presented on Morpho-MNIST, using CNNs or ResNets to classify morphed digits as in- or out-of-distribution. Additional experiments on more complex or varied datasets would strengthen the paper and help clarify when bounding yields a meaningful advantage.

**Requested Changes:**

1. The authors state “…lbMALA inherently takes [statistical dependencies between weights] into account.” Could the authors expand on what is meant by this and how lbMALA takes these dependencies into account?

2. I believe including a direct experiment of lbMALA vs MALA alone would be valuable.

3. Moreover, perhaps showing that bounding other Gaussian based methods also leads to improved performance would make the paper more convincing.

4. In the analysis of figure 4 it is stated that MC dropout performs best with CNNs on the thickening dataset, however this does not appear to be true from the plots.

5. Currently Figure 4 is very hard to see and interpret since many of the methods have similar values. I think this could be improved by for example changing the y axis range to 0.4-1.0.

Minor:
* The citations are currently incorrectly formatted which impedes readability (i.e. citep vs. citet).
* On page 3 the quotation marks around “artificial” are incorrectly formatted.
* Pg. 5 first mathematical statement: this should be more precisely stated. Currently $\epsilon$ is undefined and the probability of a sequence belonging to the typical set tends to 1 only as $n \to \infty$; the statement does not hold for small $n$.
* Algorithm 1 is missing Step 2.

---

> ### Author Response · Authors · 2025-11-14
> **Author Feedback for Reviewer pGnx**
>
> Thank you for your time and effort in reviewing our paper. We appreciate the constructive feedback and provide detailed responses below:
>
> 1. Discussion around the point (“…lbMALA inherently takes [statistical dependencies between weights] into account.” ). As the method uses an explicit posterior sampling scheme, it takes into account complex statistical dependencies between the weights. This is not a special feat of this model, but rather a property of the underlying sampling techniques. Contrast this to variational approaches that approximate the posterior with a factorized distributions that enforce independence between different weights in the posterior distribution.
> 2. The comment regarding: "The motivation and novelty are insufficiently justified". The goal here is not to get the final estimate as an average of an ensemble method. It is to estimate the uncertainty over the predictions, specifically the uncertainty coming from the uncertainty over network weights. The ensemble members provide some idea about that, and here we try to expand it further to capture the uncertainty better. We believe there is a misunderstanding and apologise for any confusion.
> 3. We thank the reviewer for noticing these discrepancies in our formula and, similar to our response to Reviewer uEJt, we noted that we have now chosen a standard notation that makes it read well and easy to follow.
> 5. All minor updates have been amended, including requested changes. Thus, in the revised manuscript, we now include a direct quantitative comparison between lbMALA and standard unbounded MALA, as requested, and have included the results in the graph of Figure 4. Furthermore, we improved the visibility and analysis to understand the overall best-performing method.

---

> > ### Comment · Reviewer_pGnx · 2025-12-03
> >
> > Thank you to the authors for their response and clarifications. After reading the revised paper, I think a lot of feedback has been taken into account and the manuscript has improved from this. Concretely, I appreciate the inclusion of a second experiment that shows the performance of the method on far OOD cases (MNIST vs FashionMNIST), and think this is an improvement, since it shows how the proposed method performs in a second different setting.
> >
> > However, I think given that the paper contribution is mainly methodological, there is still need for a more thorough benchmarking, as it is currently unclear e.g. how well the method would scale. As an example, Eschenhagen et al. [1] benchmark their method also on CIFAR-10-C and ImageNet-C, and I believe including something along these lines would be helpful. These experiments are still in the same spirit as the MNIST experiment in the paper, in that they also have varying levels of corruption, but would demonstrate that the method could scale. Further experiments could also shine light on when the method is expected to outperform other methods, since it seems that for ResNet18, lbMALA does not perform quite as well as for the CNN-based experiments.
> >
> > Moreover, the following presentational issues still occur:
> > 1. The citation styles are still incorrectly formatted. Please see https://www.jmlr.org/format/format.html#reference-section (Section Avoiding Common Errors - Citations) for details on when to use parentheses v.s. textual citations.
> > 2. Section 4.1 has introduced a spelling/formatting mistake “and bAdditionallywere used...”
> > 3. “MNIST version FashionMNIST” -> “MNIST versus …”
> > 4. Algorithm 1 line 27 “Out put” -> “Output”
> > 5. Although Figure 4 has improved, I still think this could be made more readable e.g. changing the scales of the plots, using bigger markers.
> >
> > [1] Mixtures of Laplace Approximations for Improved Post-Hoc Uncertainty in Deep Learning  https://arxiv.org/pdf/2111.03577

---

### Review · Reviewer_YSVr · 2025-10-30

**Summary Of Contributions:**

The paper introduces an improved technique for posterior sampling from high-dimensional Laplace approximations in Bayesian model averaging. The main intuition is that, in high dimensions, Bayesian models are prone to counterintuitive effects, namely, their typical set (where most random samples will fall) has almost all its mass concentrated near a thin annulus with radius $\sigma\sqrt{d}$ (assuming a Gaussian approximation $\mathcal{N}(\mathbf{0}, \sigma\mathbb{I})$). This behavior can be problematic, because posterior sampling will almost surely sample points far from the mode, and hence underrepresent the most probable weight configurations. As a result, weight configurations close to the MAP estimate will typically not be sampled by Gaussians in high dimensions. To address this, the authors propose a Metropolis-adjusted Langevin algorithm (MALA), but with a restricted domain that is close to the MAP estimate. Concretely, they define boundary conditions $d_{min}$ and $d_{max}$ and ensure that during the MALA run, all samples exceeding the boundary will reflect back. In this way, MALA still converges to a stationary distribution but within a restricted domain. In practice, a mixture of distributions in multiple high-probability regions is used, yielding a locally bounded stationary distribution in each region. Experiments on the MNIST dataset with the Morpho-MNIST toolkit for OOD detection show that the proposed method often outperforms other established competitors.

The paper is motivated by a real practical problem in posterior sampling with Gaussian approximations in high dimensions. The solution has interesting elements and the experimental evaluation shows solid performance on the MNIST dataset. On the other hand, I'm not sure all claims are supported by clear evidence. More on that in the next sections.

**Audience:**

Yes

**Audience Explanation:**

Bayesian model averaging is a well-established technique in the ML literature. The authors identify a real issue with Laplace approximations in high-dimensional settings, and convincingly argue how this can result in samples far from the MAP estimate. For this reason, I believe this work should be interesting to individuals in TMLR's audience.

**Broader Impact Concerns:**

I don't have broader impact concerns.

**Claims And Evidence:**

No

**Claims Explanation:**

I have several concerns regarding the claims made in the paper.

First, many technical details are not clear.
- The authors mention that they use the Hessian of the log-likelihood to dynamically calculate the reflective boundaries $domain_{min}$ and $domain_{max}$. How exactly do they do that? The inverse of the Hessian of the negative log-posterior at the mode corresponds to the covariance matrix of the Gaussian approximation. The authors could write down the analytical derivations to show exactly how the values are computed.
- The authors mention that "the reflective boundaries constrain the sampling to a high probability region of the posterior. On the other hand, they explain that the typical set almost exclusively falls inside a thin annulus of radius $\sigma\sqrt{d}$. But then if the two boundaries have radius $r < \sigma\sqrt{d}$, then how is the area enclosed by the two boundaries a high-probability region of the posterior? I understand the idea of forcing the samples to be within the boundary, but I wasn't clear why this region is a high-probability region of the posterior.
- The authors claim that their posterior sampling converges to a stationary distribution within the defined boundary. I was wondering if this is what really happens in practice. Does lbMALA end up reflecting back to the boundary most of the time? Or does it indeed manage to converge to a stationary distribution within the boundary? The authors could elaborate more on the method's behavior, and whether the stationary property is indeed observed empirically. For instance, one may assume that what happens in practice is that lbMALA will end up reflecting samples back to the boundary all of the time, since posterior sampling will be prone to sampling from the thin annulus where the typical set is concentrated.
- MALA should be faster than classical random-walk MCMC due to the gradient Langevin term. But I was still unclear whether the overall framework is efficient,, compared, say, to SWAG and its variants. But I'd assume it is still much slower than gradient-based Hamiltonian Monte Carlo. I feel that this aspect was not elucidated in the current draft. How fast and efficient is the proposed method? The mixing time is an important aspect of MCMC methods.

I also feel that the experimental evaluation is not particularly strong in its current form
- The MNIST experiments with the Morpho-MNIST toolkit are interesting, but the experimental study is otherwise of limited scope. For instance, I feel that near-OOD would be a very interesting setting for this paper. There are various curated datasets and benchmarks for this purpose, e.g., CIFAR-10 (ID) vs. CIFAR-100 (OOD), ImageNEt-1k (ID) vs. ImageNet-O, ImageNet-A, ImageNet-R (OOD), etc. I understand that the authors may have naturally preferred to focus on smaller and datasets, but perhaps they could have included more than just one task. Or even include far-OOD or domain-shift settings. More datasets/tasks would have provided more solid evidence for the strong performance of the proposed methods.
- Figure 4 shows overall good performance for lbLAMA, especially for the thickening OOD task, where it clearly outperforms all other methods by a good margin. But many other methods (SWAG, multi-SWAG, and even Baseline MAP)) seem to perform pretty strongly, even outperforming lbLAMA on some tasks. My impression from Figure 4 is that it shows that lbLAMA can be a good method (especially compared to MC Dropout and Laplace Mixtures), but not necessarily that it is stronger than the other well-performing methods. Perhaps a more diverse experimental evaluation would prove tis point? Furthermore, it is not clear how the complexity of the proposed method (e.g., execution time) compares to the other well-performing methods (SWAG, multi-SWAG, and even Baseline MAP)).

**Requested Changes:**

I feel some changes would be necessary for acceptance. First of all, the authors should clarify all of the points that I raised in the previous sections.
- For example, how exactly do they calculate the reflective boundaries?
- Also, why is the region enclosed by the two boundaries a high-probability posterior region, when we know that the typical set of the high-dimensional Gaussian is concentrated in a thin annulus of radius $\sigma\sqrt{d}$?
- Furthermore, how does ldMALA behave empirically? Does it for example initially do lots of reflections back to the boundary, until it converges in the space between the two boundaries? It was not clear from the text whether it even converges in the first place (empirically).
- What is the time complexity of the proposed scheme, especially compared to the other competitors? LAMA is typically considered faster than random walk-based MCMC Methods, but not as fast as other alternatives.
- What are the advantages of lbMALA compared to SWAG and its variants? They all seem to be performing quite well, and the latter may even be faster? Not saying this is the case, but it's worth clarifying this point.

I also feel that this work would be considerably strengthened by embracing a more extensive experimental study. MNIST with the Morpho-MNIST toolkit is a great starting point, but the current empirical evaluation is limited. I offered various suggestions above, but the authors may want to explore their own ideas on this,

---

> ### Author Response · Authors · 2025-11-14
> **Author Feedback for Reviewer YSVr**
>
> Thank you for your time and effort in reviewing our paper. We appreciate the constructive feedback and provide detailed responses below:
>
> 1. We have amended the wording regarding the log-posterior in the algorithm description accordingly. Additionally, the explanation of the boundary reflection was previously unclear; we have therefore derived an improved formulation that provides a more analytically sound and, in particular, updated:
> 1.1 The description regarding the negative log-posterior and the Hessian that was utilised to obtain the local curvature.
> 1.2 Clearly articulating the implementation of the reflective boundaries, including a clearer explanation of how the hyperball is formulated.
> 2. Regarding the reflective boundaries constraint, that is a very good question indeed. The thin annulus gets farther away from the posterior mode as the dimension increases. This annulus and thus the typical set are not living in an area of high likelihood. Restricting the sampling to a narrow area around the mode effectively did not let the samples get farther away from the posterior mode and stay in the high likelihood areas.
> 3. In reference to the point: "posterior sampling converges to a stationary distribution within the defined boundary." Our method targets the posterior restricted to a practically relevant region of the parameter space. By choosing the radius of the domain restriction to be larger than the radius of the Gaussian annulus, the reasoning is that no relevant probability mass lies outside of the restricted domain. Instead, this is a deliberate approximation of the area in parameter space between the Gaussian annulus and the MAP estimate. While we cannot show any experimental results, the convergence error of the full posterior consists of the "truncation error", which is the probability mass outside of the restriction, and the usual discretization error of the MALA method. While all theoretical results for the convergence of MALA still hold, the domain restriction ensures more steps of MALA in an area that includes almost all posterior mass. Our work only reports on the beneficial effects in output space.
> 4. We do not have any experimental results on the convergence itself and compared to other methods. However, the motivation for MALA was that the domain restriction is possible with this method without losing the theoretical guarantees of convergence. In other words, we choose the method because it was easily adaptable to the idea of approximating only in an area of the parameter space that we define with insights from theory about the Gaussian annulus and typical sets. Of course, it would be very interesting to see whether other posterior approximation methods can be used in this setting and what effects domain restriction has on the speed of convergence. However, this would be beyond the scope of this paper, which focuses on the evaluation of the method in output space.
> 5. Thank you for the extensive feedback regarding the boundaries, formulation, additional experiments, and for suggesting a comparison to other datasets. We note that we covered these pertinent points in a new draft of the paper. In particular, we expanded the calculations for reflections, extended the evaluation beyond Morpho-MNIST by adding Fashion-MNIST as a second benchmark dataset, and we provided evaluation times for all methods in Appendix A for more clarity.
> 6. Lastly, with regards to the point around "why is the region enclosed by the two boundaries a high-probability posterior region, when we know that the typical set of the high-dimensional Gaussian is concentrated in a thin annulus of radius". We do not claim that all points inside the bounded ball are high-probability; rather, we restrict sampling to the annular region where the posterior mass concentrates.

---

> > ### Comment · Reviewer_YSVr · 2025-11-20
> > **additional clarifications**
> >
> > I appreciate the authors' response, but I feel there are still unaddressed issues as well as various ambiguous notations.
> > - On page 5, the authors mention that almost all of the probability mass is located in a thin annulus of radius $\sigma\sqrt{d}$. Then on page 6, they define a hyperball around the $\theta_{map}$ of radius $R$, with $r < \sigma\sqrt{K}$. First, what is $R$ and $r$? Second, what is now $K$? Furthermore, in Algorithm 1 the authors define $R \gets \beta ||\sigma||_2$. Is $K$ the same as $\beta$?
> > - There were previously two different terms (min and max), whereas there is a single hyperball now. What motivated tis change?
> > - On the one hand, the authors explain that the probability mass is located in a thin annulus of radius $\sigma\sqrt{d}$. On the other hand, their defined hyperball has radius $<<\sigma\sqrt{d}$ for large $d$, since they use in practice $\beta=2$. That means that the region that they are sampling from is not of high-probability, even if it contains the mode. Did I understand something incorrectly? I understand the desire to sample from a region near the mode (as opposed the thin annulus), but I do not see how that region is high-probability based on what the authors have claimed elsewhere in the paper.
> > - On page 7, the authors mention convergence in the bounded region. To me, convergence would mean that the MALA sampling stop sampling outside of the bounded region and converges within the sampled region. But is this really what is happening? One could easily assume that the MALA sampling will continue to sample everywhere in the unbounded space, even as time approaches infinity, and the reflective condition will just keep pushing it back towards the defined hyperball. There are no arguments (theoretical and empirical) supporting convergence.
> > - Why is the performance of unbounded LAMA so inconsistent? The authors could perhaps qualitatively compare the ensemble that LAMA vs. lbLAMA come up with. For instance, for lbLAMA we know it is in the hyperball. What about unbounded LAMA? Is it empirically in the thin annulus, as the authors claim on page 5? It would be good to understand the differences.
> > - The authors did not really address my question about SWAG and its variants. Aren't these methods very comparable? What would be the benefit of using lbLAMA instead of these?
> > - Given the main contribution of this paper is the modification of an existing algorithm without any further theory, I would have expected to see more experiments. I understand the need for fine-grained OOD detection, but isn't the current benchmark limited in scope?

---

> ### Author Response · Authors · 2025-11-25
> **Feedback for additional clarifications**
>
> Thank you for your time to provide further feedback on our paper. We respond below to the points raised:
>
> Points 1 and 2: We apologise for the confusion and inconsistency. Yes, $R$ is $r$ the radius, and additionally, $K$ is the dimensionality, i.e $d$, but this has been corrected, so it is consistent now. Importantly, the scaling factor $B$ is not the dimensionality. It is our user-chosen hyperparameter that adjusts the radius of the constrained region. In our implementation, we set $B=2$ as a tuning decision, which corresponds to exploring a neighbourhood approximately within two posterior standard deviations under the Laplace approximation, which gave stability to our lbMALA simulations. Additionally, the min-max was an incorrect way to explain the hyperball because it did not explain the curvature-informed hyperball used in the code.
>
> 3. This question is directly at the heart of the counterintuitive nature of high-dimensional distributions. An important thing to note is that mass is proportional to the hyper-volume of the annulus times the likelihood. Indeed, in very high dimensions, most of the mass is located in an annulus far from the mode, and these are the areas that correspond to very low likelihood values with huge hyper-volumes. When one samples from such a distribution, the samples most likely come from the annulus. These samples correspond to low-likelihood but high mass locations due to the size of the hyper-volume. This is indeed the counter-intuitive part of high-dimensional distributions. When we restrict the samples to the hyperball, they are coming from high-likelihood areas, but these areas are also those that have less mass compared to the annulus. The fact that they are high-likelihood means they achieve low training error on the training set, which is precisely what we are aiming at.
>
> 4. The reflective boundary condition creates a stationary distribution from which MALA samples. With convergence, we refer to the convergence of the Markov chain to this stationary distribution. We attempt to explain the concept with a simplified version of the proof; thus, it can be summarised as follows. If you define a unit ball $B$ and a target density $𝜋$ supported on $B$, whose log density is differentiable with the Lipschitz continuous gradient [1], [2] and assume the reflection at the boundary dB as a Neumann boundary condition [3] then the result is a continous-time SDE which has an invariant distribution on B and satisfies the boundary condition. It can therefore be discretised and approximated by an Euler-Maruyama scheme (with reflection) to yield the reflected Langevin algorithm. [1]
>
> References:
> [1] Bubeck et al., Finite-Time Analysis of Projected Langevin Monte Carlo, NeurIPS 2015.
> [2] Boukardagha, Penalized Reflected SGLD (CFE 2024 Poster).
> [3] Lou & Ermon, Reflected Diffusion Models, PMLR 2023.
>
> 5. The inconsistency of the unbounded MALA is due to the fact that typical set of the high-dimensional posterior distribution contains samples far away from the mode. The mode is the part where the data likelihood is high - in other words, this is the location where the model works well on the training set, and thus the validation set if the model has not overfit. Sampling weight sets away from the mode can create samples from the posterior distribution, but with low data likelihood - thus low model performance. Bounding the samples guarantees in a way that the samples come from around the mode, and as a result, the resulting model will have performance similar to the mode itself.
>
> 6. We apologise for not being clearer. We see SWAG providing a fast Gaussian approximation, while lbMALA gives a curvature-aware MCMC alternative that can work directly from a MAP model and produces more stable performance across seeds and distortions, as evident from our experiments.
>
> 7. The contribution of the paper is a practical constrained MALA sampler rather than a new theoretical method; thus, our emphasis is on comparing it against the strong baselines under controlled fine-grained OOD settings. In particular, our controlled distortions allow us to evaluate the uncertainty with a higher sensitivity. The experiments we have shown indicate that lbMALA has consistent performance at similar computational costs.

---

> > ### Comment · Reviewer_YSVr · 2025-11-26
> > **thanks for clarifications**
> >
> > I thank the authors for the clarifications. There is one obvious typo on page 5 below Equation (9): $|| x - the\mu ||_2^2$ (there may be more typos, but this one stood out). Some comments/questions:
> > - I feel that the paper presentation can be confusing. I believe the authors should make clear that the region around the mode is the high probability region. But, then, due to how Gaussian distribution work in high dimensions, most of the probability mass will be concentrated in a this annulus at the given radius. Otherwise, the terms probability mass vs. high probability region can be confusing. Given this is a critical point of this work, I feel the authors should put more effort into explaining this point carefully and clearly. One  interesting idea might be through visualisations or even specific examples. For instance, explain what "high-probability mass" vs. "high-probability region" means for a specific high-D Gaussian distribution (e.g., a 20-D one).
> > - There may have been some confusion regarding my point on convergence. I agree with the authors that under specific technical assumptions, one can show concrete results. But my point was rather about how often the proposal in Algorithm 1 is accepted. Indeed, the Algorithm makes a proposal, which may be rejected or accepted. The authors do not give any information on the empirical behavior of their method. Are rejections common, and how do they affect practical convergence? I understand why the authors wanted to avoid heavy technical details, but at least they could have explored properties of their method from an empirical standpoint.
> > - On the other hand, I understand the counter-argument that "the contribution of the paper is a practical constrained MALA sampler rather than a new theoretical method", so maybe they naturally preferred to focus on comparison to other baselines. But in that case is one task sufficient to prove the value of the proposed method? If there is no theoretical contribution except a practical algorithm, perhaps it would be expected that the benefit from using the proposed framework would be shown in more benchmarks?

---

### Review · Reviewer_uEJt · 2025-11-02

**Summary Of Contributions:**

The paper proposes Mixtures of Locally Bounded Langevin Dynamics (lbMALA) as a method for Bayesian model averaging in high-dimensional neural network parameter spaces. The core idea is as follows:

- Start from an ensemble of MAP (or MAP-like) solutions (e.g., a deep ensemble).
- Around each MAP, define a fixed-radius hyperball (the radius is chosen relative to local curvature or a Laplace-based standard deviation).
- Run a Metropolis–Adjusted Langevin-type sampler constrained to remain within the hyperball via a reflective boundary condition, thereby producing a locally constrained stationary distribution.
- Combine the locally constrained posteriors from the ensemble into a mixture, sample ensemble members from these local chains, and (in implementation) fine-tune models initialized from selected samples.
- Evaluate the resulting mixture posterior using a fine-grained OOD-ranking benchmark (Morpho-MNIST distortions), and claim that lbMALA yields more reliable entropy-based ID/OOD separation than several baselines (MAP, MC dropout, SWAG, etc.).

**Strengths**:

The paper addresses an intuitively important practical problem: the typical-set phenomenon in high-dimensional posterior sampling and its effects on predictive and epistemic uncertainty estimation. The proposed hybrid between discrete-mode ensembles and locally constrained Langevin sampling is an interesting and potentially useful middle ground between point-estimate ensembles and full posterior MCMC sampling.

**Weaknesses**:

- The paper is poorly written, with several undefined or vaguely defined terms and imprecise arguments. The presentation throughout needs significant improvement for clarity and rigor.

- The manuscript uses inconsistent notation and contains factual inaccuracies, most notably in its description of the Langevin proposal and step-size conventions (see, for instance, the discussion in Section 3.4 and the unnumbered formula at the end of page 5). These must be corrected: notation of dimension suddenly changes from $d$ to $K$, curvature is undefined, etc.

- (**A Major Concern**) The motivation and novelty are insufficiently justified. Since the approach assumes access to a pre-trained ensemble of MAP estimates, the authors must clearly explain why running additional MCMC is beneficial compared to simpler alternatives such as Laplace approximations or direct ensemble averaging. In typical practice, MCMC itself is used to find MAP or high-posterior-density estimates, not to sample around already known ones.

- The theoretical justification for the reflective, locally bounded chains is thin. The paper should specify what stationary distribution these chains target, and provide at least some discussion of convergence, ergodicity, or known limitations.

- The experimental evaluation is narrowly scoped (only Morpho-MNIST). It lacks quantitative analysis of calibration, predictive log-likelihood, and computational cost. Several implementation choices such as radius selection, Hessian estimation, and hyperparameters such as step size are not justified or ablated.

The manuscript requires substantial improvements in clarity, correctness of the algorithmic formulation, theoretical grounding, and experimental validation. I recommend rejection in its current form, or at best, major revision following major rewriting and expansion of experiments.

**Additional Comments:**

- Citation formatting is inconsistent. The authors should correct their BibTeX/LaTeX usages (e.g., use `\citet` vs `\citep` consistently and ensure bracket styles are appropriate throughout the manuscript).

- The discussion of concentration of measure relies on relatively recent textbooks; please add classical foundational references (e.g., standard works on the Gaussian annulus/concentration phenomena) rather than citing only modern surveys.

- The Langevin proposal displayed at the end of page 5 is non-standard and appears incorrect. The drift/variance coefficients should follow the usual MALA parameterization (drift  ($\tau/2$), noise scale ($\sqrt{\tau}$)); please fix the formula, number the equation, and make the proposal/acceptance kernels consistent throughout the paper.

**Audience:**

No

**Audience Explanation:**

Bayesian learning of neural networks is indeed an important and broadly interesting topic within the machine learning community. However, given the current state of this manuscript: its unclear presentation, weak motivation, and insufficient experimental validation, it is unlikely to attract significant interest from TMLR’s audience in its present form.

**Broader Impact Concerns:**

No direct additional ethical concerns.

**Claims And Evidence:**

No

**Claims Explanation:**

I have the following main comments:

**Major comments**:

- The paper suffers from poor presentation and unclear definitions throughout. Many key terms, such as epistemic uncertainty, discrete/continuous support group, reflective boundary, and even $\theta_{\mathrm{MAP}}$, are either undefined or vaguely described. The writing is often confusing and hard to follow, even in the Introduction. Several sections (e.g., 3.4–3.6) use inconsistent notation and leave important quantities (e.g., curvature, Laplace approximation) unspecified. The entire paper requires substantial rewriting for clarity and rigor.

- The main motivation, that high-dimensional posteriors have “bad” typical sets far from the mode, is overstated and not convincingly argued. In practice, MCMC-based methods aggregate many samples (e.g., through averaging or more robustly via median-of-means), which mitigates this issue. The authors do not discuss why such standard remedies fail, weakening the foundation of their proposal.

- Conceptually, the approach is incoherent: if the MAP estimate (or several MAPs) is already known, it is unclear why an additional MCMC phase is needed. MCMC is typically used to find or approximate MAP or posterior expectations, not to sample around an already-known mode. The authors should clarify the purpose of local Langevin sampling in this context and why it improves over simpler alternatives like Laplace or ensemble averaging.

**Minor comments**:

- The description of MALA and the transition kernel $q(\cdot)$ is incomplete and inconsistent across sections. The Langevin proposal formula is not in its standard form; the paper should present it correctly and define all related quantities.

- Experimental evidence (limited to Morpho-MNIST) is too narrow to support general claims, and no ablation or calibration metrics are provided.

**Requested Changes:**

See the comments.

---

> ### Author Response · Authors · 2025-11-14
> **Author Feedback for Reviewer uEJt**
>
> Thank you for your time and effort in reviewing our paper. We appreciate the constructive feedback and provide detailed responses below:
>
> 1. We appreciate the concerns regarding unclear terminology and inconsistent notation. We agree that greater alignment is necessary to ensure theoretical coherence. Therefore, we have revised the flow and updated the equations to provide improved clarity and a more consistent narrative throughout the paper, including the update of citation formats.
> 2. To the points about unclear definitions: Epistemic uncertainty has now been updated in the first two lines of the Introduction. The remaining points we believe regarding the discrete/continuous support group are well defined within the context of the paper and the standpoint from which it is written. However, should the reviewer require further derivations, we can discuss and include them in a future revision.
> 3. We thank the reviewer for your insights and major concerns and would like to offer some feedback. First of all, an important note here is that this work focuses on uncertainty and not the averaging or median of means approaches for extracting a single prediction. We would like to quantify the variation in the prediction. The current methods argue that they gather posterior samples in the weight space to explain the variance, in particular, the epistemic uncertainty. If most of these samples are coming from typical sets, where the data likelihood is indeed low, the value of the corresponding epistemic uncertainty computation becomes questionable. Indeed, we know many many weight sets will yield high accuracy over a training set, in other words, will have equally high data likelihood. It is the variation in prediction arising from these weight sets that we should be interested in, not the variation seen in weight samples from the typical sets. Lastly, using the average of samples from the typical set is the approach of rather ignoring the variation and thus not truly quantifying the epistemic uncertainty. The problem arising from averaging low-likelihood models to yield a final prediction is also evident. These points demonstrate why ensemble methods are currently undisputed champions in quantifying epistemic uncertainty and robust predictions.
> 4. The contribution of the Langevin sampling around modes, which are extracted by ensembling, is to effectively increase the diversity of the samples beyond the modes. If the modes coming from the ensembling capture a part of the total diversity, sampling around the modes aims to capture a larger part. An important distinction we would like to make here is that "averaging" is never really the aim of this work. The aim is to quantify the diversity and the variability of the prediction arising from the diversity of the weight sets that can equally well explain the training data. One can, however, question whether to use a Laplace approximation or a Langevin sampling around the modes. That is precisely the experimental analysis provided in the article. We believe that, taking into account the local shape of the posterior distribution, Langevin sampling should capture the posterior distribution around the mode better.
> 5. We thank the reviewer for noticing the discrepancies in the transition kernel and Langevin proposal. Similar to our response earlier, we noted that we have now chosen a standard notation that makes it read well and easy to follow.
> 6. In the revised manuscript, we extend the evaluation beyond Morpho-MNIST by adding Fashion-MNIST as a second benchmark dataset, following the reviewer’s recommendation for broader empirical validation, and also report on run times in Appendix A.
> 7. We thank the reviewer for the suggestion to strengthen the discussion of concentration of measure by including more classical foundational references. In addition to the recent textbooks previously cited, we have now incorporated the following reference:
> 7.1. Regarding the typical set first introduced in the paper in the Introduction (p. 2), we cite a classical information theory from MacKay’s Information Theory, Inference, and Learning Algorithms (MacKay, 2003). This reference offers one of the clearest and earliest descriptions of typical-set behaviour and provides valuable foundational context.
> 7.2. Second, for the Gaussian annulus and concentration results, we note that the presentation in Foundations of Data Science by Blum, Hopcroft, and Kannan (particularly Chapter 2) provides a standard and mathematically rigorous statement of the Gaussian Annulus Theorem. We have therefore not added any other reference regarding this point. We believe this text sufficiently covers the foundational material requested by the reviewer.
> 8. Lastly, an additional sampling phase is needed because MAP only gives a point estimate, but including local Langevin sampling will capture the surrounding posterior uncertainty, which Ensembles can not achieve, nor Laplace, as it enforces a Gaussian approximation.

---

> > ### Comment · Reviewer_uEJt · 2025-11-30
> >
> > I would like to thank the authors for their response. I have read the authors’ reply, reviewed the other reviewers’ comments and discussions, and examined the revised manuscript. Most of my concerns regarding notation inconsistencies and lack of clear explanations are largely addressed, and I believe the paper is now more readable. Nevertheless, I suggest a more thorough rewriting to further improve clarity.
> >
> > My main concern, however, remains unresolved. I still fail to understand the primary motivation behind the proposed approach. Even if the authors aim to capture epistemic uncertainty and are concerned that noisy samples may fall in low-likelihood regions (per the Gaussian annulus theorem), why not simply use a smaller $\tau$ (or, in the paper’s notation, $\epsilon$)? For instance, could one choose an $\epsilon$ that scales proportionally to $d^{-1/2}$?
> >
> > Moreover, the contribution of projecting Langevin dynamics samples into a ball of limited radius to bring them closer to the MAP estimates appears fairly incremental.

---

### Author Response · Authors · 2025-11-14
**lbMALA Author Feedback**

Good day, We would like to thank the reviewers for their time and effort in providing insightful and valuable feedback. We have summarised our responses below and made corresponding updates to the paper in light of all comments received. We address each reviewer’s feedback separately in each official comment. We apologise for the delay, as we thought that we had 14 days to respond and make changes.

Lastly, we corrected some of our diagrams that had the wrong headings, and we hope that we have sufficiently covered reviewers' concerns and look forward to the feedback.

Thank you so much and keep well.

---

### Author Response · Authors · 2025-12-05
**Delay in Responses**

Good day All,

I hope all is well.

We apologise for the delay in responses. Unfortunately, the main responder has been ill this week but will endeavour to provide feedback in the next few days.

Thank you in advance for your understanding, and again, my apologies for the delay.

Keep well.

---

> ### Comment · Editors_In_Chief · 2025-12-19
>
> Hi there, just wanted to check in. Will there be responses soon?

---

> > ### Author Response · Authors · 2025-12-19
> > **General Feedback and Comments**
> >
> > Good day,
> >
> > Firstly, our apologies for the delay as we have been wrapping up a few items.
> >
> > We want to reply to all the latest comments in a consolidated message to note:
> >
> > 1. We have carefully considered the reviewers’ comments and will incorporate additional experiments on CIFAR-10 and CIFAR-10C into the paper for all methods and architectures. Completing these experiments will require additional time, and we therefore kindly request that all changes and amendments be finalised by early February, as multiple parties are involved and prior commitments need to be coordinated. This date provides a large safety net considering the festive period ahead.
> > 2. Lastly, we thank the reviewers for their careful reading of the revised manuscript and for the constructive feedback each reviewer gave us. We are also pleased that the concerns regarding notation consistency and clarity have largely been addressed. We do appreciate the suggestion for further improving the paper in terms of readability and further clarifications, and will continue refining the presentation to enhance clarity throughout the paper.
> >
> > Thus, if the proposed date incorporating all the above changes is suitable, we will continue working on the paper and ensure that the deadline is met.
> >
> > Thank you very much for your time and consideration.

---

> > > ### Comment · Action_Editor_oNEB · 2025-12-23
> > >
> > > Dear Authors,
> > >
> > > Thank you for your response. The proposed period is acceptable. We will consider 13 February as the new hard deadline for submitting the revised version of your manuscript.
> > >
> > > Yours sincerely,
> > > AE

---

> > > > ### Author Response · Authors · 2025-12-23
> > > > **Noted and Appreciated**
> > > >
> > > > Good day,
> > > >
> > > > Thank you for the confirmation and for your consideration regarding the final date. We sincerely appreciate it.
> > > >
> > > > Take care, and we wish you a wonderful festive season.
> > > >
> > > > Keep well.

---

### Author Response · Authors · 2026-02-12
**Final submission - Consolidating All Reviews**

Good day,

Thank you for your reviews. I hope you are well and off to a good start to the New Year.

We believe we have now incorporated and addressed all previously raised points. We have included a few final comments highlighted below, and the revised version of the manuscript has been uploaded.

Reviewer YSVr: Comment regarding paper presentation.
Response: We agree with the reviewer, and we should make clear that the region around the mode is the high probability region. We have now amended the paper (Please see Section 3.2) to make a distinction early on in the article between regions where the probability density function achieves a high value and regions where random samples are most likely to come from, i.e., high probability mass regions. In a 20D Gaussian, the region that achieves the highest probability density function value is the mode of the Gaussian. However, the samples are most likely to come from a region far from the mode. A vague analogy can be made to mass = volume x density. Consider a 20D unit Gaussian centered at the origin. Now, consider a set of concentric hyper-spherical shells with
 thickness starting at the origin and emanating outwards. The innermost shell will be at the mode of the Gaussian distribution, hence the density value will be very high. However, the volume of that shell will be very small, hence the mass will be relatively small. A shell that is very far away will have a very large volume but will reside in an area with very low density, thus it will also have a low mass. A shell that is in between will strike the right balance. The density at that region times the volume of that shell will yield the highest mass. Similarly, in the 20D Gaussian, samples will come from that shell that strikes a good balance between volume and density value, not the shell that is at the highest density area, but with a very small volume.

Reviewer uEJt:
1. Comment on the primary motivation and motivation.
Response: Choosing a smaller epsilon may seem to address the issue at first; however, unfortunately, it does not. Decreasing epsilon will yield samples from the Markov Chain that are too similar to each other. In other words, the autocorrelation of samples in the Markov Chain increases as epsilon decreases. The samples will not cover the posterior and thus will lead to an underestimation of the actual variability, in our case, of the epistemic uncertainty. Here, our approach restricts the posterior distribution but still allows the model to move with longer steps within that area to select samples. The selected samples, therefore, are less similar to each other compared to using a smaller $epsilon.

2. Comment on the contribution of projecting Langevin dynamics samples into a ball of limited radius.
Response: To the best of our knowledge, this insight has not been presented before. It looks at epistemic uncertainty estimation methods for neural networks from a very different angle and questions the validity of these techniques, explaining their unintuitive behavior. It goes further than simply identifying a problem and presents a practical solution. Therefore, we respectfully disagree with the reviewer. The final method may seem incremental, but we believe it should be considered within the context of the entire article.

We believe the changes to the paper and comments above should now adequately resolve all outstanding items.

Thank you so much for your time and patience.

Keep well!

---

> ### Comment · Reviewer_uEJt · 2026-02-25
> **Acknowledgment of Response**
>
> Thank you for your clarifications.
>
> Based on the authors’ response, my concern regarding the choice of not setting $\epsilon \propto d^{-1/2}$ has been resolved. However, I encourage the authors to include this justification in the main body of the paper as well.

---

> > ### Author Response · Authors · 2026-02-25
> > **Acknowledgment of Reviewer uEJt's Feedback**
> >
> > Good Morning,
> >
> > We hope all is well.
> >
> > Thank you to Reviewer uEJt for your response. We shall incorporate the aforementioned details into the paper.
> >
> > Keep well, and thank you for all the input and help.

---

### Comment · Action_Editor_oNEB · 2026-04-28
**Camera Ready Version**

Dear Authors,

Please upload the camera ready version of your paper at your earliest convenience.

Yours,

AE

---

> ### Author Response · Authors · 2026-04-28
> **Query - Camera Ready Version**
>
> Good day,
>
> Thank you, we just testing the repo and then should upload today.
>
> I wanted to kindly check the link to be updated in latex for the camera ready document will be this one: https://openreview.net/forum?id=5860 as the ID is the paper ID correct?
>
> Thank you so much!

---

> > ### Comment · Action_Editor_oNEB · 2026-04-28
> >
> > No it is usually not
> >
> > AE

---

### Comment · Editors_In_Chief · 2026-06-02

On June 2, by request of the authors, the EiCs replaced the camera ready version with a version including acknowledgments.

---

### Decision · Action_Editor_oNEB · 2025-12-03

**Recommendation:** Accept as is

**Audience:**

Yes

**Audience Explanation:**

Langevin Dynamics and Bayesian Model Averaging are of substantial interest to the TMLR community

**Claims And Evidence:**

Yes

**Claims Explanation:**

The revised manuscript addresses several substantive concerns raised in the initial round. In particular, the presentation has been improved, the conceptual motivation is clearer, and the revision adds useful empirical evidence, including the CIFAR-10/CIFAR-10-C benchmark and additional information on sampler behavior such as cumulative acceptance rates. These changes make the paper significantly easier to assess and strengthen the overall case for the proposed approach.

The main remaining concern, reflected in multiple reviews, is that the empirical advantage of lbMALA over standard MALA and other baselines is not uniformly strong across all settings. In several experiments, lbMALA is competitive rather than clearly dominant, and some competing methods outperform it on particular distortions or benchmarks. That said, the revised experimental evidence does support the conclusion that lbMALA is a strong and generally reliable method overall, with favorable average performance and a coherent motivating framework.

Taking the reviews and revision together, my assessment is that the paper meets the standard for publication. While the empirical gains are not consistent enough to support a stronger endorsement, the revised manuscript is sufficiently clear, technically motivated, and empirically supported to warrant acceptance.

---

> ### Author Response · Authors · 2026-03-24
> **Timeline Negotiation for TMLR**
>
> Good day,
>
> Thank you so much for the positive feedback and response to our updates.
>
> I wanted to kindly check if it would be possible to extend the deadline to complete the camera-ready version to the end of April? I will be on maternity leave in May and have a few things to finish, and would appreciate the slight movement in date.
>
> I hope this will be possible and apologise for any trouble - if I am done earlier, it will definitely be submitted before the end of April too.
>
> Thank you so much and keep well.

---

> > ### Comment · Action_Editor_oNEB · 2026-03-24
> >
> > Of course, and congratulations!
> >
> > Yours,
> >
> > AE

---

> > > ### Author Response · Authors · 2026-03-24
> > > **Reply Timeline Negotiation for TMLR**
> > >
> > > Evening,
> > >
> > > Thank you so much for the kind consideration and wishes.
> > >
> > > Take good care!